# Mutual interaction between visual homeostatic plasticity and sleep in adult humans

Danilo Menicucci[1]*, Claudia Lunghi[2], Andrea Zaccaro[1], Maria Concetta Morrone[3,4], Angelo Gemignani[1]

[1]Department of Surgical, Medical and Molecular Pathology and Critical Care Medicine, University of Pisa, Pisa, Italy; [2]Laboratoire des Systèmes Perceptifs, Département d'études Cognitives, École Normale Supérieure, Paris, France; [3]Department of Translational Research and New Technologies in Medicine and Surgery, University of Pisa, Pisa, Italy; [4]IRCCS Fondazione Stella Maris, Pisa, Italy

**Abstract** Sleep and plasticity are highly interrelated, as sleep slow oscillations and sleep spindles are associated with consolidation of Hebbian-based processes. However, in adult humans, visual cortical plasticity is mainly sustained by homeostatic mechanisms, for which the role of sleep is still largely unknown. Here, we demonstrate that non-REM sleep stabilizes homeostatic plasticity of ocular dominance induced in adult humans by short-term monocular deprivation: the counterintuitive and otherwise transient boost of the deprived eye was preserved at the morning awakening (>6 hr after deprivation). Subjects exhibiting a stronger boost of the deprived eye after sleep had increased sleep spindle density in frontopolar electrodes, suggesting the involvement of distributed processes. Crucially, the individual susceptibility to visual homeostatic plasticity soon after deprivation correlated with the changes in sleep slow oscillations and spindle power in occipital sites, consistent with a modulation in early occipital visual cortex.

*For correspondence: danilo.menicucci@unipi.it

Competing interest: The authors declare that no competing interests exist.

## Editor's evaluation

Menicucci and colleagues investigated the potential role of sleep in the homeostatic plasticity of ocular dominance in adult humans. This is a careful study that should be of broad interest to those studying adult cortical plasticity, particularly in vision. The study shows that sleep can maintain the changes in ocular dominance obtained after applying an eye patch on the dominant eye for two hours, which contrasts with the rapid decline of these changes during quiet wake in darkness. The authors further report correlations between sleep oscillations and the magnitude of the plasticity effect. Overall, these results implicate sleep in a new form of plasticity.

## Introduction

Neural plasticity is an intrinsic property of the nervous system underlying the ability to change in response to environmental pressure. Learning and memory processes plastically and continuously encode new neuronal information during wakefulness, but consolidation mechanisms often extend into sleep (*Rasch and Born, 2013*). During non-REM (NREM) sleep, replay, and pruning processes as well as connectivity rearrangements associated with specific neuronal activity patterns involving GABAergic modulation (*Ma et al., 2018*) play a fundamental role in plasticity consolidation.

Two hallmark rhythms characterize the background of on-going oscillations of NREM sleep: slow wave activity (SWA, 0.5–4 Hz) and sigma band ($\sigma$, 9–15 Hz). A fundamental contribution to the lower

frequency bound of SWA is due to sleep slow oscillations (SSOs), an EEG pattern that corresponds to the alternation between periods of neuronal membrane depolarization and sustained firing (upstates) and periods of membrane hyperpolarization and electrical silence (downstates) whereas power in the sigma rhythm is provided by the sleep spindles: waxing and waning wave packages that spread throughout the thalamocortical system (*Steriade, 2006*). SSOs and sleep spindles have been consistently associated with synaptic plasticity, replay, and memory consolidation (*Rasch and Born, 2013*; *Crunelli et al., 2018*; *Antony et al., 2018*). SSOs allow spike-timing-dependent-plasticity, while replay occurs via cortico-thalamo-cortical interactions that are made effective through thalamus-cortical synchronization in the sigma band: the sleep spindle is the full-fledged expression of this mechanism (*Capone et al., 2019*). Several studies have investigated the role of slow and spindle oscillations in episodic and procedural memory (*Holz et al., 2012*; *Miyamoto et al., 2017*), while the contribution of these processes in sensory plasticity has yet to be assessed.

Growing evidence indicates that the adult human visual system might retain a higher degree of plasticity than previously thought (*Castaldi et al., 2020*; *Baroncelli and Lunghi, 2021*). Some degree of Hebbian plasticity is retained in adulthood and mediates visual perceptual learning (*Watanabe and Sasaki, 2015*) as well as visuo-motor learning (*Huber et al., 2004*; *Menicucci et al., 2020*). The residual Hebbian plasticity in adult involves changes in neural processing occurring at multiple levels of visual (perceptual learning *Dosher and Lu, 2017*) and visuo-motor (*Ghilardi et al., 2000*) processing, particularly at associative cortical level, and are consolidated by both NREM (*Huber et al., 2004*; *Menicucci et al., 2020*) and REM (*Boyce et al., 2017*) sleep. While sleep-dependent Hebbian plasticity has been shown in thalamocortical circuits in rodents visual system following prolonged exposure to a novel visual stimulus (*Durkin et al., 2017*), Hebbian plasticity of ocular-dominance in the primary visual cortex (V1) is very weak or absent in primates after closure of visual critical period.

Ocular dominance plasticity (*Espinosa and Stryker, 2012*; *Wiesel and Hubel, 1963*; *Hubel and Wiesel, 1970*) is an established model of sensory plasticity in V1 in vivo, which is observed after a period of monocular deprivation (MD) (*Hubel and Wiesel, 1970*; *Berardi et al., 2000*). Ocular dominance plasticity is maximal during development, when it is mediated both by Hebbian and by homeostatic plasticity, which have different functional outcomes. Hebbian plasticity stabilizes the most successful inputs in driving neural activity, and consequently the deprived eye loses the ability to drive cortical neurons (*Cooke and Bear, 2014*). On the other hand, homeostatic plasticity upregulates the neuronal response gain of the weakened deprived eye. During development, Hebbian and homeostatic mechanisms work hand in hand and their relative strength depends on timing and MD duration (*Kaneko and Stryker, 2017*; *Turrigiano, 2017*). The homeostatic ocular dominance plasticity is preserved through the life-span: in adult humans, recent studies have shown that short-term MD (2–2.5 hr) unexpectedly shifts ocular dominance in favour of the deprived eye (*Lunghi et al., 2011*; *Lunghi et al., 2013*; *Zhou et al., 2013*). This counterintuitive result, consistent with homeostatic plasticity, is interpreted as a compensatory adjustment of contrast gain in response to deprivation. The deprived eye boost is observed also at the neural level, as revealed by EEG (*Lunghi et al., 2015a*), MEG (*Chadnova et al., 2017*), and fMRI (*Binda et al., 2018*; *Kurzawski et al., 2022*) and, importantly, it is mediated by a decrease of GABAergic inhibition in the V1 (*Lunghi et al., 2015b*). The effect of short-term MD decays within 90 min from eye-patch removal (*Lunghi et al., 2011*; *Zhou et al., 2013*). However, recent evidence from a clinical study in adult patients with anisometropic amblyopia shows that repeated short-term deprivation of the amblyopic eye can promote the long-term recovery of both visual acuity and stereopsis (*Lunghi et al., 2019b*), suggesting that the effect of short-term MD can be consolidated over time. In adult amblyopic patients, the classic occlusion therapy (*Webber and Wood, 2005*), which consists in the long-term deprivation of the non-amblyopic eye and relies on Hebbian mechanisms, is much less effective than the inverse occlusion approach (234 hr of traditional occlusion per 1 line of visual acuity improvement [*Fronius et al., 2014*] vs. 12 hr of inverse occlusion per 1.5 lines of visual acuity improvement [*Lunghi et al., 2019b*]). Inverse occlusion involves the short-term deprivation of the amblyopic eye, relies on the homeostatic plasticity mechanisms described above and is consolidated for up to 1 year (*Lunghi et al., 2019b*). Understanding the role of sleep for the maintenance of visual homeostatic plasticity induced by short-term MD is therefore a clinically relevant and timely question.

Evidence from animal models shows that NREM sleep is necessary to consolidate Hebbian mechanisms during ocular dominance plasticity within the critical period (*Aton et al., 2009*; *Durkin and*

*Aton, 2019*; *Aton et al., 2013*). However, it is still largely unknown whether sleep has similar effects after the closure of the critical period in the visual cortex. In addition, it is still unknown whether sleep can modulate homeostatic plasticity induced by MD (*Lunghi et al., 2011*; *Lunghi et al., 2013*; *Zhou et al., 2013*; *Chadnova et al., 2017*; *Binda et al., 2018*; *Bai et al., 2017*; *Lunghi and Sale, 2015c*; *Binda and Lunghi, 2017*; *Lunghi et al., 2019a*; *Zhou et al., 2015*; *Ramamurthy and Blaser, 2018*).

There are several mechanisms that are shared between homeostatic plasticity and sleep, indicating a possible interaction between the two phenomena. Homeostatic plasticity is based on GABA-dependent mechanisms (*Desai et al., 2002*; *Maffei et al., 2010*) that alter the excitation/inhibition balance and appear to be analogous to the factors that modulate the expression of slow-wave sleep (*Luppi et al., 2017*). More generally, plasticity induced by learning does change local cortical mechanisms that are stabilized by sleep, and the stabilization relies upon hippocampal activity during NREM in several experimental models including sensory memory (*Ji and Wilson, 2007*; *Preston and Eichenbaum, 2013*; *Sigurdsson and Duvarci, 2015*; *Yamada, 2022*). The potential hippocampal involvement can be traced by measuring sleep spindles that invade the cortex from this structure during NREM sleep in order to support memory consolidation.

Here, we investigate the effect of NREM sleep on visual homeostatic plasticity in adult humans, both analysing sleep features at the occipital and the prefrontal cortex levels.

## Results

We assessed visual cortical plasticity by measuring the effect of short-term (2 hr) of MD on ocular dominance measured by binocular rivalry (BR) (*Levelt, 1967*; *Alais and Blake, 2005*; *Blake and Logothetis, 2002*) in adult volunteers. In the experimental night (monocular deprivation night [MDnight]), MD was performed in the late evening and was followed by 2 hr of sleep, during which high-density EEG was recorded (*Figure 1A*). At this night awakening, ocular dominance was assessed again, and then participants went back to sleep until the morning (4–5 hr of additional sleep). For the control condition (control night [Cnight]), the same participants underwent an identical protocol, but without performing MD (*Figure 1B*).

Consistently with previous reports (*Lunghi et al., 2011*; *Lunghi et al., 2013*), short-term MD shifted ocular dominance in favour of the deprived eye (*Figure 1C*, red symbols, repeated-measures ANOVA, $F(4,72)=6.7$, $p<0.001$, $\eta 2=0.27$): the deprivation index (DI) was significantly altered just after eye-patch removal (mean DI ± SE = 0.77±0.04, two-tailed, one sample t-test $t(18) = -5.41$, $p\_fdr = 0.00017$, Cohen's d=1.24).

Importantly, this form of homeostatic plasticity was maintained after 2 hr of sleep (mean DI ± SE = 0.87±0.04, two-tailed, one sample t-test $t(18)=-3.54$, $p\_fdr = 0.0046$, Cohen's d=0.81) and during the first 8 min after morning awakening (mean DI ± SE = 0.91±0.04, two-tailed, one sample t-test $t(18)=-2.55$, $p\_fdr = 0.02$, Cohen's d=0.59), that is about 6–7 hr after eye-patch removal. All those measurements are referred to the baseline condition taken just before the deprivation. No consistent changes in ocular dominance were observed in the control night (*Figure 1C*, black symbols, repeated-measures ANOVA, $F(4,68)=0.97$, $p=0.43$, $\eta 2=0.05$), when calculating the same index for measurement at the same time but without MD.

Exploratory post hoc analyses revealed however that in the control night the DI measured before sleep, compared to the same measurement performed 2 hr before, was significantly larger than one (mean DI ± SE = 1.08±0.03, paired-samples t-test $t(17)=-2.7$, $p\_fdr = 0.045$, Cohen's d=0.62), favouring the non-dominant eye. This might reflect a transient form of adaptation or implicit learning due to the repetition of the BR test (*Klink et al., 2010*), suggesting that the effect of MD may be overall underestimated.

That the effect of deprivation is maintained for several hours after eye-patch removal is surprising, because the effect of short-term MD normally decays within 90 min (see also *Figure 2—figure supplement 1*; *Turrigiano, 2017*; *Lunghi et al., 2013*). Our results therefore indicate that sleep maintained visual homeostatic plasticity, stabilizing the ocular dominance change induced by MD and delaying the expected decay until the awakening. Interestingly, the effect of MD (DI) measured before and after 2 hr of sleep did not correlate across subjects (Spearman's rho = 0.18, p=0.47). This suggests that individual sleep pattern could interact with visual homeostatic plasticity and that the instatement and maintenance of plasticity might be mediated by different neural processes, possibly reflected in different features of NREM sleep.

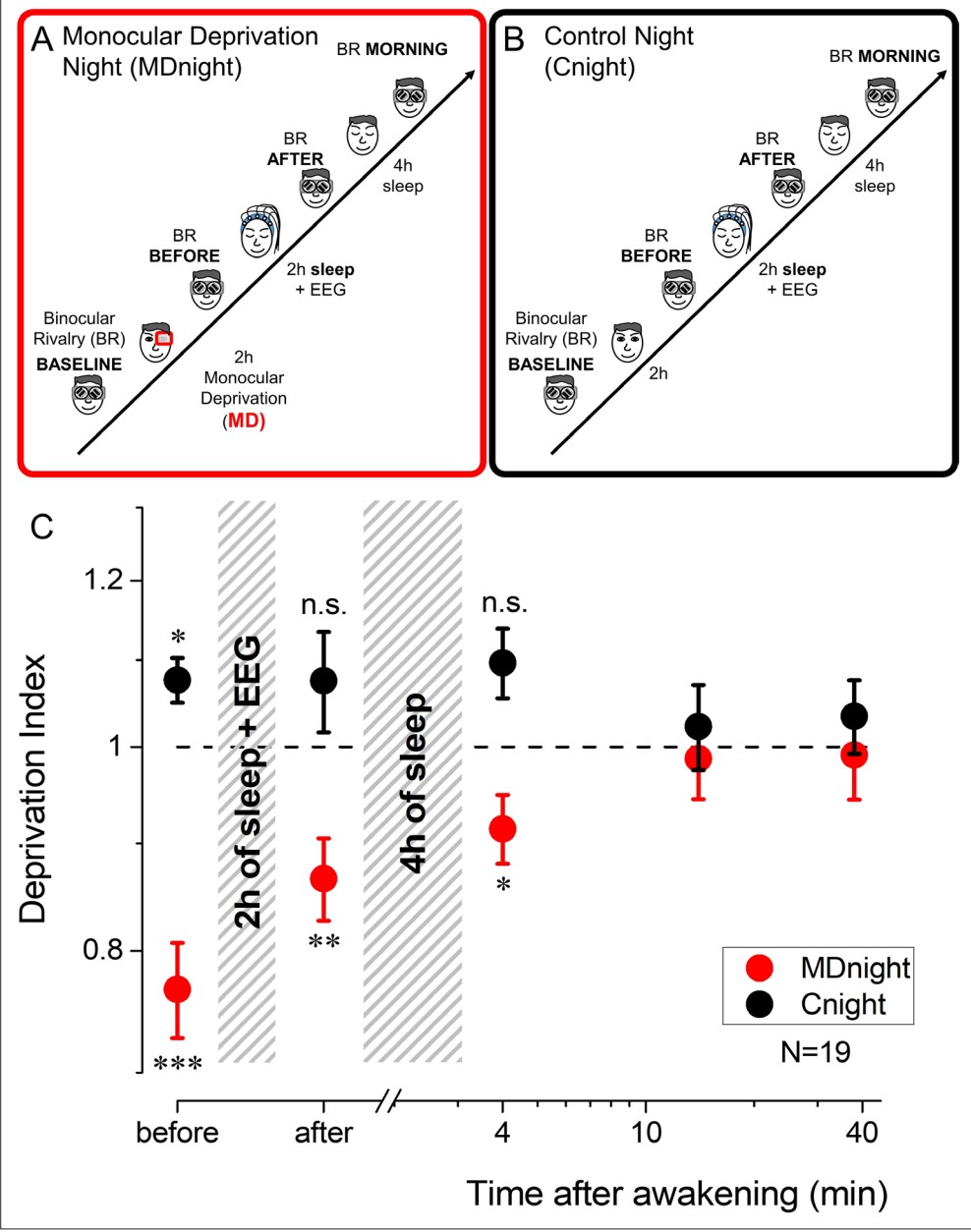

**Figure 1.** Experimental paradigm and monocular deprivation (MD) effect before and after sleep. (**A**) Diagram of the experimental paradigm for the MD night (MDnight) condition. Ocular dominance was measured by means of binocular rivalry (BR) before and after 2 hr of MD. Afterwards, participants went to sleep while their EEG activity was recorded with a 128-electrodes system. BR was measured after 2 hr of sleep and in the morning, after 4 additional hours of sleep. (**B**) Same as A, but for the control night (Cnight). The experimental procedure was the same as the MDnight, except that participants did not underwent MD. (**C**) The MD effect (deprivation index) measured before, after 2 hr of sleep and at morning awakening, occurring after 4 additional hours of sleep, in the MDnight (red symbols) and Cnight (black symbols). N=19. Error bars represent 1 ± SEM. Asterisks indicate the significance level (t-test of individual time-points against the value 1) after correction for multiple comparisons: ***p<0.001, **p<0.01, *p<0.05 or non-significant (n.s.).

The online version of this article includes the following source data for figure 1:

**Source data 1.** Source data for the monocular deprivation effect before and after sleep.

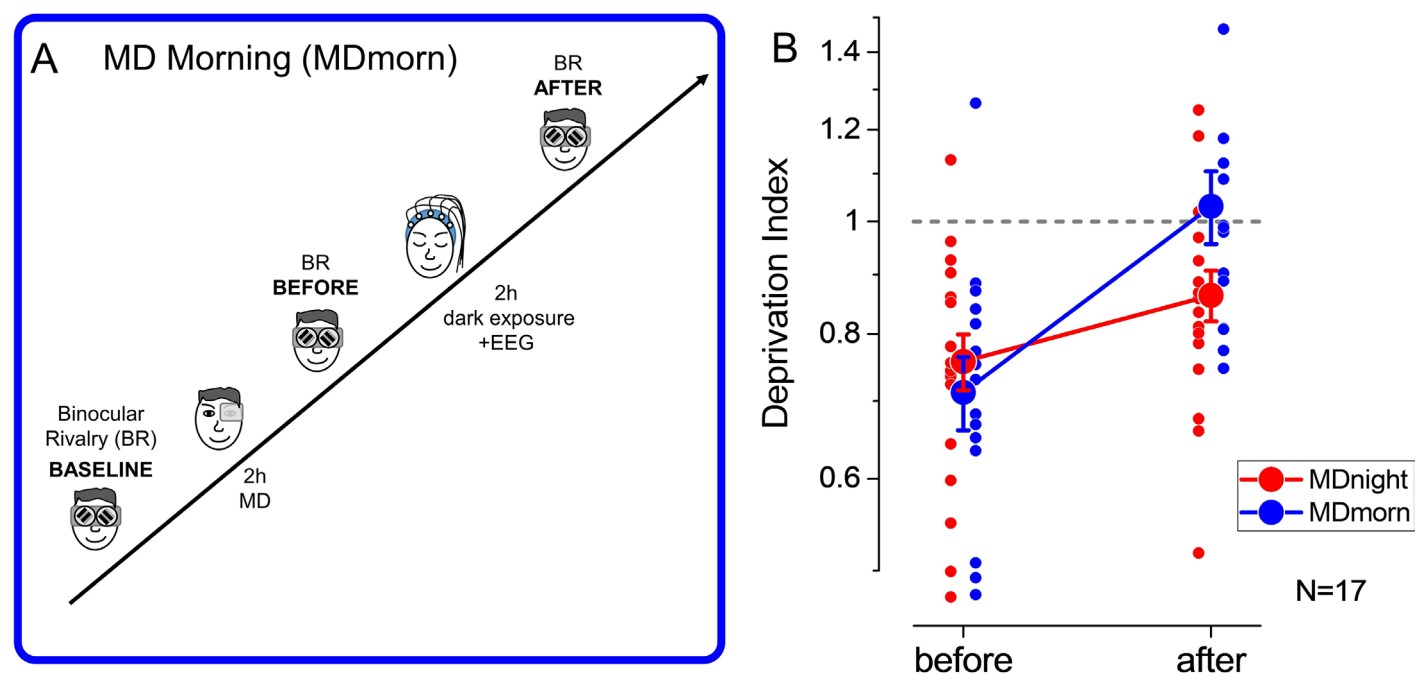

**Figure 2.** Experimental paradigm and results of the dark exposure (monocular deprivation morning [MDmorn]) condition. (**A**) Diagram of the experimental paradigm for the monocular deprivation (MD) morning condition. After 2 hr of MD, participants spent 2 hr in total darkness, while their EEG activity was recorded. Following the 2 hr of dark exposure, ocular dominance was assessed by binocular rivalry. (**B**) The deprivation index measured before and after 2 hr of dark exposure without sleep performed in the MD morning session (blue symbols) and before and after 2 hr of sleep in the MD night session (red symbols), for the 17 participants who performed both conditions. Error bars represent 1 ± SEM, the small dots represent individual subjects, the big dots represent the average.

The online version of this article includes the following source data and figure supplement(s) for figure 2:

**Source data 1.** Source data for the monocular deprivation effect before and after sleep/dark exposure.

**Figure supplement 1.** Control experiment results.

**Figure supplement 1—source data 1.** Source data for the control experiment.

To rule out the effect of total darkness exposure occurring during sleep, we performed an additional condition during which participants, after MD, laid down in a completely dark room for 2 hr, without sleeping (monocular deprivation morning, MDmorn, *Figure 2A*). The experiment was performed in the morning, to prevent the occurrence of sleep during the 2 hr of dark exposure. In this experiment (*Figure 2B*, blue symbols), we found that the effect of MD (mean DI ± SE = 0.71±0.05, two-tailed, one-sample t-test, t(16)=–4.91, p<0.0002, Cohen's d=1.19) decayed to baseline within 2 hr of darkness (mean DI ± SE = 1.03±0.07, two-tailed, one-sample t-test, t(16) = –0.04, p=0.96, Cohen's d=0.009), similarly to what observed with normal visual stimulation after patch removal. We then directly compared the decay of the effect of deprivation after 2 hr of sleep (*Figure 2B*, red symbols) or after 2 hr of dark exposure (*Figure 2B*, blue symbols) by performing a 2 (TIME, before and after) × 2 (CONDITION, MDnight and MDmorn) repeated-measures ANOVA. We found a significant interaction between the factors CONDITION and TIME (repeated-measures ANOVA, F(1,16) = 4.48, p=0.05, $\eta$2=0.22), confirming the specific role of sleep in stabilizing visual homeostatic plasticity induced by MD.

The 2 hr of sleep and dark exposure took place at different times of the day, one late at night and the other one early in the morning. To rule out a possible influence of the circadian rhythm on homeostatic plasticity and its decay, we performed a control experiment in which we measured the effect of 2 hr of MD early in the morning or late at night in a separate group of adult volunteers. We found that the dynamics of the MD effect were similar for the morning and evening sessions, as in both cases ocular dominance returned to baseline levels within 120 min (repeated-measures ANOVA, TIME*CONDITION: F(4,32) = 1.08, p=0.38, $\eta$2=0.12). Moreover, the MD effect was significantly

larger when deprivation was performed in the morning (*Figure 2—figure supplement 1*, repeated-measures ANOVA, CONDITION: F(1,8)=6.87, p=0.031, $\eta$2=0.46), indicating a lower plastic potential of the visual cortex in the evening. Taken together, these results indicate that the maintenance of the MD effect is specific to sleep. Moreover, the lower plastic potential observed in the evening indicates that the role of NREM sleep in stabilizing homeostatic ocular dominance plasticity might be underestimated.

Having demonstrated a stabilization of the boost of the deprived eye with sleep, two different but intermingled issues arise: (1) how MD affects subsequent sleep, and (2) how sleep contributes to stabilize visual homeostatic plasticity. As the first 2 hr of night sleep contained none or just a few minutes of REM sleep, both effects have been investigated within NREM sleep.

NREM sleep features were derived from the 2 hr of EEG recording before the first night awakening. These include the power scalp distribution of SWA (0.5–4 Hz) and sigma (9–15 Hz) rhythms, the rate (waves per time unit) and shape of SSO as well as the density (waves per time unit) and power of sleep spindles.

Table in *Supplementary file 1* shows descriptive statistics of sleep architecture parameters in the MDnight and Cnight. Also, it provides between-nights statistical comparisons and the study of putative association between sleep architecture, susceptibility to MD (DI before) and stability of the effect during sleep (DI after). No sleep architecture parameters varied significantly between nights or were associated with DI measurements.

Given previous evidence of ocular dominance homeostatic plasticity at level of early visual cortex (*Binda et al., 2018*), we analysed the EEG rhythms and patterns in an extended occipital ROI reflecting the activity of the majority of primary and associative visual areas. We compared the changes over the ROI from the control to experimental night of each feature. A control ROI was selected in correspondence of the sensory-motor cortex (*Figure 3—figure supplement 1*).

Large ROIs have the advantage to compensate for the individual large variations of source dipole localization across visual areas and to decrease uncorrelated neuronal noise by averaging. We also performed single electrode analysis obtaining similar, although noisier, results (see scalp maps in *Figures 3 and 4*, and single electrode correlations distributions in *Figure 3—figure supplement 2* and *Figure 4—figure supplement 1*).

We never observed any main changes of the sleep rhythms or features between MDnight and Cnight recordings, in any of the ROIs (Table in *Supplementary file 2*). However, we observed a strong correlation between changes in sleep features and ocular dominance plasticity measured before sleep in the occipital ROI (*Figure 3* and *Figure 3—figure supplement 2* for the coherence of correlations between electrodes within ROIs).

For the SSOs, a strong correlation with plasticity was observed between changes in SSO rate (rho = −0.66, p=0.009, $p_{fdr}$ = 0.036, *Figure 3, A*) and shape (slope+, rho = −0.64, p=0.012, $p_{fdr}$ = 0.037, *Figure 3, B*; SSO amplitude, rho = –0.56, p=0.03, $p_{fdr}$ = NS, *Figure 3—figure supplement 3, A*) at the level of the occipital ROI: subjects showing a stronger plasticity effect showed (1) increased SSO rate, (2) increased sharpness of transitions from the SSO negative peak, whereas subjects with lower plasticity effect exhibited opposite parameters changes.

The opposite changes between subjects with high and low plasticity, without any main effect of MD manipulation on SSO features, suggest different local synchronization in the sleeping neurons undergoing the bistability behaviour of SSO. Finally, for the sleep spindles, a correlation with plasticity was observed between their mean power (rho = −0.66, p=0.009, $p_{fdr}$ = 0.036; *Figure 3C*): power of sleep spindles over the occipital sites increased in the MDnight for participants showing higher visual homeostatic plasticity (i.e. a large boost of the deprived eye), while the power was reduced in participants showing a lower response to MD. This result, together with the other sigma power estimates, reinforces the idea behind a modulation of thalamocortical interaction as a homeostatic reaction to MD. Indeed, sigma activity power expressed during the whole NREM sleep in the occipital ROI correlated with individual ocular dominance shifts (rho = −0.66, p=0.009, $p_{fdr}$ = 0.036, *Figure 3—figure supplement 3, B*) and overlapping result was observed when considering the sigma rhythm expressed just before SSO events that favour the emergence of full-fledged SSO (rho = −0.70, p=0.0046, $p_{fdr}$ = 0.025, *Figure 3—figure supplement 3, C*). These strong correlations contrast with the absence of association with the power of SWA during the whole NREM sleep with the plasticity index (*Figure 3—figure supplement 3, D*).

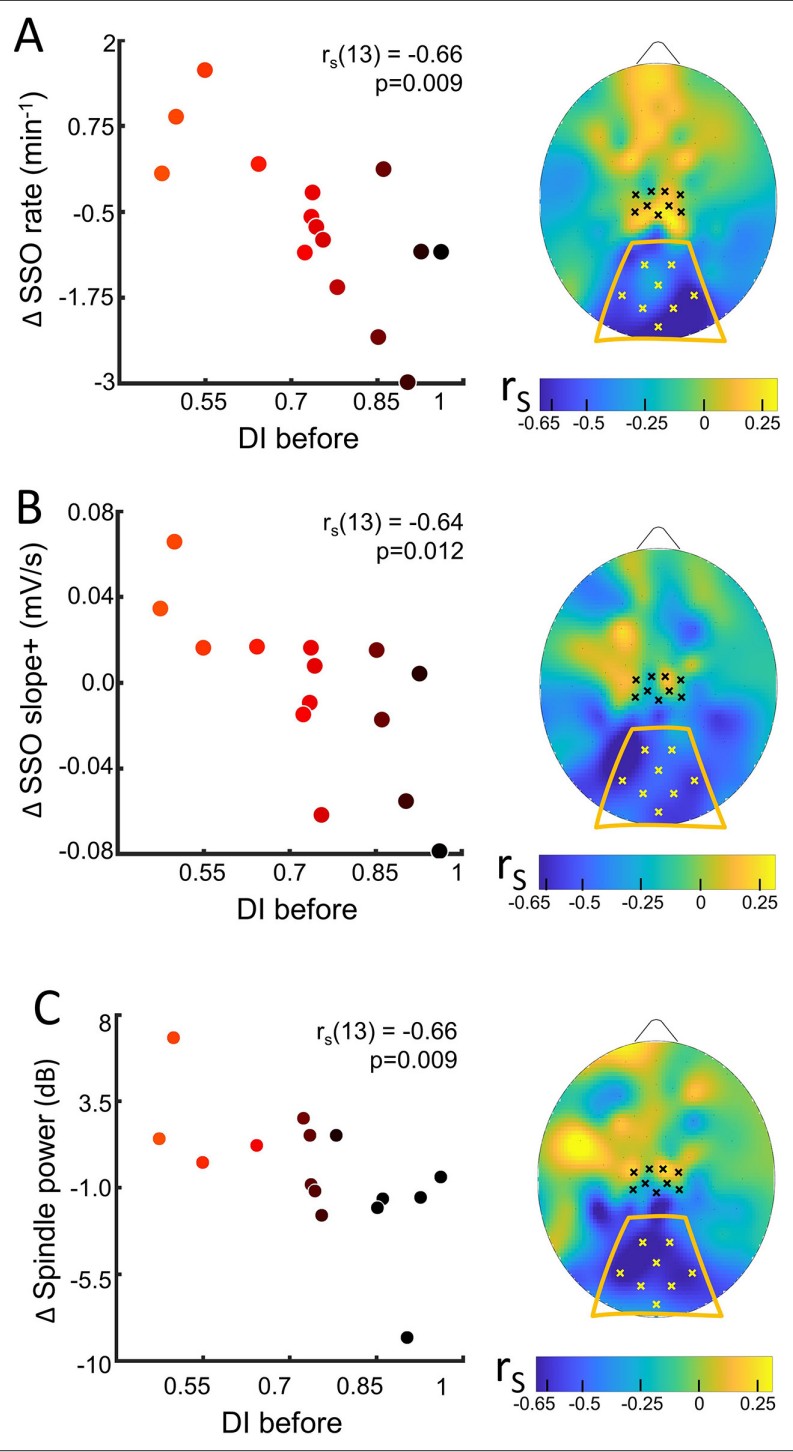

**Figure 3.** Sleep slow oscillation (SSO) and spindle modulation with the eye dominance induced by monocular deprivation. (**A**) The changes from control to MD night of the rate of SSOs was correlated (N=15) with the deprivation index measured before sleep (DI before). Scatterplot shows individual values averaged within the occipital ROI (Spearman's r with the p-value shown as an inset in the scatterplot). Colour of dots spans from black to red as a function of individual plasticity. No significant correlation appeared when considering the control ROI defined in the sensory-motor cortex. The scalp map shows the spatial distribution of correlation as estimated electrode by electrode; within the map, yellow and black dots mark electrodes belonging to the occipital and sensory-motor ROIs, respectively. (**B**) Same as A for the steepness of slope+ of SSOs; (**C**) same as A for the spindle power.

*Figure 3 continued on next page*

*Figure 3 continued*

The online version of this article includes the following source data and figure supplement(s) for figure 3:

**Source data 1.** Source data for the scatterplot of the sleep slow oscillations (SSOs) rate averaged over the occipital ROI versus the deprivation index (DI before).

**Source data 2.** Source data for the scatterplot of the sleep slow oscillation (SSO) slope+ averaged over the occipital ROI versus the deprivation index (DI before).

**Source data 3.** Source data for the scatterplot of the spindle power averaged over the occipital ROI versus the deprivation index (DI before).

**Source data 4.** Source data for the scalp map of the correlations between sleep slow oscillations (SSOs) rate and the deprivation index (DI before).

**Source data 5.** Source data for the scalp map of the correlations between sleep slow oscillation (SSO) slope+ and the deprivation index (DI before).

**Source data 6.** Source data for the scalp map of the correlations between spindle power and the deprivation index (DI before).

**Figure supplement 1.** Electrodes in the HydroCel Geodesic Sensor Net belonging to each ROI.

**Figure supplement 1—source data 1.** Source data for the EEG electrode coordinates.

**Figure supplement 2.** Correlations (N=15) with deprivation index (DI) before sleep.

**Figure supplement 2—source data 1.** Source data for the single electrode correlations within each ROI between sleep slow oscillation (SSO) rate and the deprivation index (DI) before sleep.

**Figure supplement 2—source data 2.** Source data for the single electrode correlations within each ROI between sleep slow oscillation (SSO) slope+ and the deprivation index (DI) before sleep.

**Figure supplement 2—source data 3.** Source data for the single electrode correlations within each ROI between spindle power and the deprivation index (DI) before sleep.

**Figure supplement 3.** Sleep slow oscillation (SSO) and sigma power modulation with the eye dominance induced by monocular deprivation.

**Figure supplement 3—source data 1.** Source data for the scatterplot of the sleep slow oscillations (SSOs) amplitude averaged over the occipital ROI versus the deprivation index (DI before).

**Figure supplement 3—source data 2.** Source data for the scatterplot of the sigma activity power averaged over the occipital ROI versus the deprivation index (DI before).

**Figure supplement 3—source data 3.** Source data for the scatterplot of the sigma rhythm expressed before sleep slow oscillation (SSO) events averaged over the occipital ROI versus the deprivation index (DI before).

**Figure supplement 3—source data 4.** Source data for the scatterplot of the power of slow wave activity averaged over the occipital ROI versus the deprivation index (DI before).

**Figure supplement 3—source data 5.** Source data for the scalp map of the correlations between sleep slow oscillations (SSOs) amplitude and the deprivation index (DI before).

**Figure supplement 3—source data 6.** Source data for the scalp map of the correlations between sigma activity power and the deprivation index (DI before).

**Figure supplement 3—source data 7.** Source data for the scalp map of the correlations between sigma rhythm expressed before sleep slow oscillation (SSO) events and the deprivation index (DI before).

**Figure supplement 3—source data 8.** Source data for the scalp map of the correlations between slow wave activity and the deprivation index (DI before).

**Figure supplement 4.** Monocular deprivation in the morning session (MDmorn): maps of Spearman's correlations (N=15) calculated between EEG power band content and deprivation index before and after 2 hr of dark exposure.

**Figure supplement 4—source data 1.** Source data for the scalp maps of the correlations between bands power and the deprivation index before the 2 hr of dark exposure (DI before).

**Figure supplement 4—source data 2.** Source data for the scalp maps of the correlations between bands power and the deprivation index after 2 hr of dark exposure (DI after).

---

The observed significant correlations were coherent at the single electrode level within the occipital ROI (*Figure 3—figure supplement 2*), but they were not observed in the sensory-motor ROI or at the electrode level within this control ROI. Also they were specific for sleep as no correlation between EEG rhythms power and visual plasticity was observed in the control experiment (darkness exposure condition, *Figure 3—figure supplement 4*).

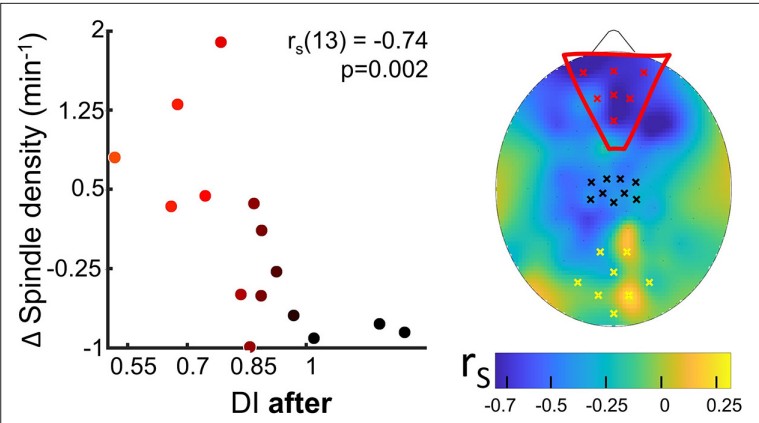

**Figure 4.** Spindle density modulation with the residual plasticity after 2 hr of sleep. The changes from control to MD night of the spindle density were correlated (N=15, p<0.05, FDR-corrected) with the deprivation index measured after sleep (DI after). Scatterplot shows individual values averaged within the prefrontal ROI (Spearman's r with the p-value shown as an inset in the scatterplot). Colour of dots spans from black to red as a function of individual DI after. No significant correlation appeared when considering ROIs defined in the occipital and in sensory-motor cortex. The scalp map shows the spatial distribution of correlation as estimated electrode by electrode over the scalp; within the map yellow, black, and red dots mark electrodes belonging to the occipital, sensory-motor, and prefrontal ROIs, respectively.

The online version of this article includes the following source data and figure supplement(s) for figure 4:

**Source data 1.** Source data for the scatterplot of spindle density averaged over the prefrontal ROI versus the deprivation index (DI after).

**Source data 2.** Source data for the scalp map of the correlations between spindle density and the deprivation index (DI after).

**Figure supplement 1.** Correlations with deprivation index (DI) after sleep.

**Figure supplement 1—source data 1.** Source data for the single electrode correlations within each ROI and the deprivation index (DI) after sleep.

**Figure supplement 2.** Sleep slow oscillation (SSO) and sigma power modulation with the residual plasticity after 2h of sleep.

**Figure supplement 2—source data 1.** Source data for the scatterplot of the power of slow wave activity averaged over the prefrontal ROI versus the deprivation index (DI after).

**Figure supplement 2—source data 2.** Source data for the scatterplot of the power of sigma power averaged over the prefrontal ROI versus the deprivation index (DI after).

**Figure supplement 2—source data 3.** Source data for the scatterplot of the power of sleep slow oscillation (SSO) rate averaged over the prefrontal ROI versus the deprivation index (DI after).

**Figure supplement 2—source data 4.** Source data for the scatterplot of the power of sleep slow oscillation (SSO) slope+ power averaged over the prefrontal ROI versus the deprivation index (DI after).

**Figure supplement 2—source data 5.** Source data for the scalp map of the correlations between sigma power and the deprivation index (DI after).

**Figure supplement 2—source data 6.** Source data for the scalp map of the correlations between slow wave activity and the deprivation index (DI after).

**Figure supplement 2—source data 7.** Source data for the scalp map of the correlations between sleep slow oscillation (SSO) rate and the deprivation index (DI after).

**Figure supplement 2—source data 8.** Source data for the scalp map of the correlations between sleep slow oscillation (SSO) slope+ and the deprivation index (DI after).

**Figure supplement 3.** Across sleep analysis.

**Figure supplement 3—source data 1.** Source data for the scalp map of the correlations between changes within the sleep cycle in sleep slow oscillation (SSO) amplitude and the deprivation index (DI after).

**Figure supplement 3—source data 2.** Source data for the scalp map of the correlations between changes within the sleep cycle in spindle power within the sleep cycle and the deprivation index (DI after).

*Figure 4 continued on next page*

*Figure 4 continued*

**Figure supplement 3—source data 3.** Source data for the scalp map of the correlations between changes within the sleep cycle in sleep slow oscillation (SSO) slope+ and the deprivation index (DI after).

**Figure supplement 3—source data 4.** Source data for the scalp map of the correlations between changes within the sleep cycle in the sigma rhythm expressed before sleep slow oscillation (SSO) events and the deprivation index (DI after).

To investigate how sleep contributes to the maintenance of the acquired ocular dominance, we also analysed sleep features in frontal electrodes. We correlated ocular dominance plasticity measured after night awakening (*Figure 1A*, *BR after*) with sleep characteristics in the three target ROIs measured in the early night: the occipital, the control sensory-motor, and a new prefrontal ROIs (*Figure 4* and *Figure 4—figure supplement 1* for the coherence of correlations across electrodes within each ROI).

Sleep spindles density in the prefrontal ROI strongly correlated (rho = −0.74, p=0.002, $p_{fdr}$ = 0.04) with the effect of MD as maintained after sleep (*Figure 4*): participants retaining the stronger effect after 2 hr of sleep showed substantial increase of spindle density. No correlation was observed in the other ROIs. Besides, neither power of SWA and sigma bands, nor SSO rate and shape, were associated with the residual eye dominance after 2 hr of sleep (*Figure 4—figure supplement 2*).

Finally, we considered the variations of EEG parameters during the 2 hr of sleep to verify the stability of the sleep features as a function of the delay from deprivation. Among the considered EEG measures (*Figure 4—figure supplement 3*), none was associated with the residual eye dominance after 2 hr of sleep (DI after).

## Discussion

We investigated the interplay between visual homeostatic plasticity and sleep in healthy adult humans after a short MD period. We report for the first time that different features of NREM sleep affect and are affected by homeostatic ocular dominance plasticity in adult humans: the plastic potential of the visual cortex is reflected by the expression of SSO and sigma activity in occipital areas and sleep consolidates the effect of short-term MD via increased spindles density in prefrontal area.

### Role of sleep in supporting visual plasticity

We found that sleep promotes the stabilization of visual homeostatic plasticity induced by 2 hr of MD: the deprived eye boost observed after MD, which normally decays within 90 min of wakefulness (*Lunghi et al., 2013*; *Zhou et al., 2013*; *Lunghi et al., 2011*), is maintained by sleep for up to 6 hr after eye-patch removal. Importantly, this effect was specifically induced by sleep: 2 hr of dark exposure after MD did not prevent the decay of the effect. Finally, we found that the circadian rhythm did not influence the dynamics of the effect, as a similar time course of plasticity was observed when MD was performed early in the morning or late at night.

Results from animal models have shown that NREM sleep consolidates ocular dominance plasticity during the critical period (*Aton et al., 2009*; *Durkin and Aton, 2019*; *Aton et al., 2013*). This sleep-dependent consolidation relies on Hebbian mechanisms and it is mediated by synaptic potentiation through increased NMDAR and PKA activity (*Aton et al., 2009*) and decreased GABAergic inhibition (*Aton et al., 2013*), potentiating the input that more successfully drives the output activity. This type of Hebbian plasticity has never been observed in ocular dominance plasticity in adult humans. On the other hand, homeostatic plasticity, the capacity to upregulate the gain of the weak input, has been consistently reported. Here, we show for the first time that NREM sleep has an effect on ocular dominance plasticity also past the critical period in humans.

The sleep homeostatic hypothesis (*Tononi and Cirelli, 2014*) should not be confused with the homeostatic process that drives this type of plasticity. For OD the homeostatic mechanism acts on the overall activity both for the deprived and non-deprived eye, while the sleep homeostatic hypothesis states that during NREM sleep, synaptic weights only down-scale in compensation for neural activity during wakefulness (*Tononi and Cirelli, 2014*). While the theory is validated by a large set of experimental findings, it may not be easily directly linked to our results.

The homeostatic plasticity process triggered by MD is associated with up-regulation of the deprived eye activity, but also with down-regulation of the non-deprived eye at the level of the V1

(*Lunghi et al., 2015a*; *Chadnova et al., 2017*; *Binda et al., 2018*; *Zhou et al., 2015*). NREM sleep appears therefore to intervene in a rebalancing process involving overall visual cortical activity. This rebalancing process operates in opposite directions for the deprived and non-deprived eye, which explains the lack of an association between sleep features at the occipital level and the maintenance of the MD effect observed after sleep as well as the lack of a main effect of MD on the subsequent sleep features.

Interestingly, we found that the influence of sleep on ocular dominance plasticity was reflected by some sleep features in high-level areas, indicating a role of extra-striate cortex in regulating visual plasticity in adults. The density of sleep spindles was associated with the MD effect maintained after sleep in a distinct cluster of electrodes distributed over the mesial frontal and prefrontal cortex. Sleep spindles have been described as replay events of new information acquired during wakefulness and in the course of consolidation during NREM sleep, which might mark phasic activations of a circuit involving the hippocampus (*Born, 2010*): indeed the consolidation of several forms of plasticity has been associated with sleep spindle density (*Clemens et al., 2005*; *Gais et al., 2002*) and some forms of visual plasticity (e.g. perceptual learning) are reflected in changed activity and connectivity within the hippocampus (*Urner et al., 2013*), which has dense anatomical connections with the V1 (*Maller et al., 2019*). This is a surprising results given the large evidence that the mechanisms underlying ocular dominance are in V1. However, we cannot exclude that the V1 may be under hippocampal control in wakefulness, as visual memory may require replay activity. Interestingly, simple orientation texture discrimination tasks are also stabilized during sleep and also in these case frontal activity are implicated in the stabilization process (*Yamada, 2022*).

Altogether, consistent with other results, we show that activity in non-visual areas plays a role in modulating the decay of short-term visual plasticity in adult humans and that this activity might be crucial to promote the stabilization of the plastic changes induced by MD that we observed in ambly-opic patients (*Lunghi et al., 2019b*). In amblyopia, the boosting effect after MD was stabilized across consecutive days with sleep occurring in-between sessions and became permanent after performing short-term MD over 4 weeks.

## Sleep and the plastic potential of visual cortex

We found that the expression of SSO in visual areas reflects the interindividual variability in visual homeostatic plasticity: their rate increased and their shape changed in occipital sites proportionally to the shift in ocular dominance induced by MD, as measured immediately before sleep.

Less SSOs observed in low plasticity subjects also means less downstate periods. During the downstate a large majority of neurons are silent for fractions of second and restorative processes occurring at the level of individual brain cells occur (*Vyazovskiy and Harris, 2013*). Thus, less SSOs could indicate an average lower need of such processes due to the previous sensory deprivation. In this line, also healthy volunteers who sleep after blindfolding exhibit a dramatic decrease of SSOs (*Korf et al., 2017*). At the other end, more SSOs in subjects with high visual plasticity could indicate the homeo-static activation of this mechanism endorsing the ocular dominance shift. SSO shape changed accordingly after MD: subjects with high visual plasticity show greater SSOs with steeper downstate exit slope (slope+) compared to their basal characteristics. Larger SSOs indicate larger groups of cortical neurons synchronously involved in these bistable events, while a steeper positive slope has been associated with a stronger coupling with thalamic structures (*Gemignani et al., 2012*; *Esser et al., 2007*).

Altered activity in the slow wave frequency band (the band including SSOs) was also observed in sleep subsequent sensorimotor deprivation (*Huber et al., 2006*), however SSO events were not studied directly in this experimental model. The type of plasticity induced by sensorimotor deprivation differs from the effect of short-term MD, consistently with Hebbian mechanisms mediating sensory-motor plasticity and homeostatic mechanisms mediating short-term ocular dominance plasticity. Short-term MD induces opposite changes in VEPs, increasing VEPs amplitude of the deprived eye and decreasing that of the non-deprived eye (*Lunghi et al., 2015a*), while sensorimotor deprivation reduces SEPs/MEPs related to the immobilized arm leaving unaltered SEPs/MEPs related to the free arm (*Huber et al., 2004*). These differences might explain the different pattern of sleep changes observed after sensorimotor deprivation and after MD: while *Huber et al., 2006*, reported a significant decrease in SWA power, we did not observe a main effect of our experimental manipulation, reflecting balanced overall activity changes of the deprived and non-deprived eye between the pre- and post-deprivation

state. However, we reported a strong correlation between SSO rate/shape parameters and the MD effect, suggesting that SSO encodes homeostatic plasticity in the adult visual cortex.

The expression of oscillating activity in the sigma band also changed as a function of the individual plastic potential of the visual cortex. Most of the sigma activity during NREM sleep relies on the pacemaker cells within the reticular thalamic nucleus forcing patterned activity in thalamocortical and cortical cells. The paradigmatic expression of this activity is the sleep spindle during which ideal conditions for fine scale plasticity occur: thalamic inputs during spindles yield to dendritic depolarization but keeping the cell from firing and triggering calcium entry into dendrites (*Sejnowski and Destexhe, 2000*). Recordings in vivo from lateral geniculate nucleus and visual cortex have shown that SSOs and sigma activity are strongly coordinated within thalamocortical circuits (*Contreras and Steriade, 1995*), and evidence are accumulating for reticular thalamic plasticity sustaining spindle-based neocortical–hippocampal communication (*Durkin et al., 2017*). Therefore, during NREM sleep, any factor modulating the activity of one of the two structures affects both and this lays the foundation for coordinated cortico-thalamic plastic changes (*Crunelli et al., 2018*; *Gemignani et al., 2012*; *Crunelli and Hughes, 2010*; *Jaepel et al., 2017*; *Krahe and Guido, 2011*).

Overall, these results suggest that SSOs and sigma activity reflect the degree of homeostatic plasticity induced by short-term MD. Both homeostatic plasticity (*Desai et al., 2002*; *Maffei and Turrigiano, 2008*) and SSO (*Luppi et al., 2017*; *Sanchez-Vives et al., 2010*) are linked to GABAergic inhibition, the observed effect could therefore in principle be mediated by a change in excitation/inhibition balance in the visual system. Importantly, in humans, GABA concentration measured after 2 hr of MD in V1 decreased proportionally to the observed ocular dominance shift (*Lunghi et al., 2015b*), and SWA expression increases in response to GABA agonists administration (*Faulhaber et al., 1997*). Interestingly, a recent study (*Tamaki et al., 2020*) reported that complementary changes in the excitation/inhibition balance measured in the visual cortex of adult humans during NREM and REM sleep are correlated with visual plasticity induced by perceptual learning, further pointing to a leading role of GABAergic inhibition in mediating these two phenomena. We speculate that the alteration of SSO shape and density observed at the occipital level during the sleep immediately following MD might reflect the change in GABA concentration induced by deprivation and be a neurophysiological marker of the interindividual variability in the level of plasticity in adult humans.

## Conclusions

Sleep oscillatory activity can reflect the plastic potential of the occipital cortex: people highly susceptible to a visual manipulation (short-term MD) affecting V1 activity show increased SSO and sigma activity in occipital sites. Sleep can also extend for many hours an otherwise transient unbalance of visual cortical activity, and sleep spindles in prefrontal regions appear to support the process as subjects exhibiting stronger maintenance had increased frontopolar sleep spindle density, as it occurs for many memory consolidation processes.

## Materials and methods
### Participants

Nineteen healthy volunteers (mean age ± SD 24.8±3.7 years, range 21–33 years; 8 males), participated in the main study. The eligibility of each volunteer was verified by semi-structured interviews conducted by a senior physician and psychiatrist (AG) based on the following inclusion criteria: no history of psychiatric/neurological disorders (including sleep disorders), being drug free for at least 1 month. All subjects had normal or corrected-to-normal visual acuity (ETDRS charts). Enrolled volunteers received the following instructions to be accomplished in each day of the experimental procedures: to avoid any alcohol intake, coffee intake in the evening before sleep sessions and physical workout just before all the experimental sessions (both the night and the morning sessions). All subjects performed the three experimental conditions, except for one subject, who did not perform the control night condition, and two subjects who did not perform the MD morning condition, because of personal problems. Four participants were excluded from the EEG analysis because poor signal quality in the EEG signal occurred during the sleep recording.

Sample size was determined based on previous studies on sleep and plasticity (e.g. visual *Tamaki et al., 2020*; *Censor et al., 2006*; *Karni et al., 1994* and visuo-motor learning *Huber et al., 2004*;

*Ngo et al., 2013*; *Marshall et al., 2006*) using a within-subject design, indicating that robust results can be observed with a sample size of 10–15 participants. In addition, since this is the first study exploring the role of sleep in homeostatic plasticity induced by MD and that for its nature it has to take into account the complexity of sleep as expressed through different parameters, this sample size and the need for multiple tests corrections have led to highlight only effects characterized by strong effect size.

Nine additional volunteers (mean age ± SD 25.7±5.2 years, range 20–37 years; 3 males) participated in the control experiment. All participants had normal or corrected-to-normal visual acuity (ETDRS charts) and no history of psychiatric/neurological disorders.

*Source code 1* include program codes used to produce statistics and related outputs.

## Monocular deprivation

MD was performed by applying a custom-made eye-patch on the dominant eye. Eye dominance was defined according to the BR measurement performed in the baseline (11 right eyes were deprived in the main experiment and 5 right eyes in the control experiment). The eye-patch was made of a translucent plastic material that allows light to reach the retina (attenuation 15%) but completely prevents pattern vision. MD lasted 2 hr, during which participants stayed in the laboratory control room under experimenters' supervision and did activities such as reading and working on the computer. In the last half hour all subjects underwent the EEG montage so that at the patch removal, after the acquired eye dominance measurement, they could go to sleep without any delay.

## Binocular rivalry

### Main study

Visual stimuli were generated by the ViSaGe stimulus generator (CRS, Cambridge Research Systems), housed in a PC (Dell) and controlled by Matlab programs. Visual stimuli were two Gaussian-vignetted sinusoidal gratings (Gabor Patches), oriented either 45° clockwise or counterclockwise (size: 2s=2°, spatial frequency: 2 cpd, contrast: 50%) displayed on a linearized 20 inch Clinton Monoray (Richardson Electronics Ltd., LaFox, IL) monochrome monitor, driven at a resolution of 1024×600 pixels, with a refresh rate of 120 Hz. To facilitate dichoptic fusion stimuli were presented on a uniform grey background (luminance: 37.4 cd/m$^2$, CIE: 0.442 0.537) in central vision with a central black fixation point and a common squared frame. Subjects received the visual stimuli sitting at the distance of 57 cm from the display through CRS Ferro-Magnetic shutter goggles that occluded alternately one of the two eyes each frame.

Each BR experimental block lasted 3 min. For each block, after an acoustic signal (beep), the BR stimuli appeared. Subjects reported their perception (clockwise, counter clockwise, or mixed) by continuously pressing with the right-hand one of three keys (left, right, and down arrows) of the computer keyboard. At each experimental block, the orientation associated to each eye was randomly varied so that neither subject nor experimenter knew which stimulus was associated with which eye until the end of the session, when it was verified visually.

### Control experiment

The visual stimuli were generated in Matlab (R2020b, The MathWorks Inc, Natick, MA) using Psychtoolbox-3 (Brainard, 1997) running on a PC (Alienware Aurora R8, Alienware Corporation, Miami, FL) and a NVIDIA graphics card (GeForce RTX2080, Nvidia Corporation, Santa Clara, CA). Visual stimuli were presented dichoptically through a custom-built mirror stereoscope and each subject's head was stabilized with a forehead and chin rest positioned 57 cm from the screen. Visual stimuli were two sinusoidal gratings oriented either 45° clockwise or counterclockwise (size: 2°, spatial frequency: 2 cpd, contrast: 50%), presented on a uniform grey background (luminance: 148 cd/m$^2$, CIE x=0.294, y=0.316) in central vision with a central white fixation point and a common squared white frame to facilitate dichoptic fusion. The stimuli were displayed on an LCD monitor (BenQ XL2420Z 1920×1080 pixels, 144 Hz refresh rate, Taipei, Taiwan). Each BR experimental block lasted 3 min. For each block, after an acoustic signal (beep), the BR stimuli appeared. Subjects reported their perception (clockwise, counterclockwise, or mixed) by continuously pressing with the right-hand one of three keys (left, right, and down arrows) of the computer keyboard. At each experimental block, the orientation associated with each eye was randomly varied.

## High-density EEG recordings

EEG was recorded using a Net Amps 300 system (Electrical Geodesic Inc, Eugene, OR) with a 128-electrodes HydroCel Geodesic Sensor Net. The EEG system employs a high spatial density electrode system with full head coverage including periocular, cheekbone, and neck sensors. This allows the detection of both vertical and horizontal eye movements and muscle tone (both from the zygomaticus major muscles and upper trapezius muscles on the neck) directly from the EEG cap. During the 2 hr of recordings, electrode impedances were kept below 50 KΩ and signals were acquired with a sampling rate of 500 Hz, using the Electrical Geodesic Net Station software, Version 4.4.2. EEG recordings were analysed using tailored codes written in Matlab (MathWorks, Natick, MA) and EEGLAB toolbox functions (*Delorme and Makeig, 2004*).

## Experimental procedures

### Main study

Experimental procedure comprises three sessions, and for each volunteer, sessions were completed within a month and at least 1 week apart. The experiment took place in a dark and quiet room, with a comfortable bed equipped for EEG recordings and the apparatus for measuring BR placed next to the bed. Each volunteer spent two nights (from 9:30 PM to 8 AM) and a morning (from 9 AM to 2 AM) at the laboratory: (1) an MDnight, in which participants underwent 2 hr of MD before sleep; (2) a Cnight, in which no MD was performed before sleep, but participants waited 2 hr in the laboratory performing the same activities and undergoing eye dominance measures at the same times as in MDnight; (3) an MDmorn, during which subjects, after the MD, lied in the same bed of night sessions, resting in the dark for 2 hr, avoiding sleeping, in order to study the acquired eye dominance extinction without visual stimuli. The order of the night sessions was randomized and balanced within the experimental group.

For the night sessions (MDnight and Cnight), the volunteer was welcomed in the laboratory around 9.30 PM and then tested for the baseline measurement of BR. Consequently, in the MDnight session around 10 PM the monocular patch was applied, the volunteer spent 2 hr reading or watching movies and thus, after exactly 2 hr, the patch was removed. The volunteer, already wearing the EEG cap, was tested again for the second measure of BR. To calculate the DI (see *Equation 1*) before sleep we used the first and second measure of BR. Once the second BR measurement was carried out, the subject was immediately invited to go to bed to fall asleep, this happened around midnight for both MDnight and Cnight. During both nights, sleep was interrupted after 2 hr (around 2.30 AM) to test the volunteer again for the third measure of BR (it took about 10 min), after which volunteers could sleep undisturbed until 7.30 AM (from 4 to 5 hr).

For assessing ocular dominance during night sessions, BR was measured at four different times: before MD (or waiting for the Cnight, night *baseline*, 2×3 min blocks), after 2 hr of MD and before sleep (or waiting for the Cnight, *before sleep*, 2×3 min blocks), after the first 2 hr of sleep (*after sleep*, 2×3 min blocks) and after the second awakening (*morning awakening*, 5×3 min blocks measured 0, 5, 10, 15, and 30 min after eye-patch removal). Similarly, during the MDmorn session, BR was measured at three times: before MD (morning *baseline*, 2×3 min blocks), after 2 hr of MD and before dark exposure (*before dark*, 2×3 min blocks), after the 2 hr dark exposure (*after dark*, 2×3 min blocks).

During the night sessions, EEG was acquired from the in-bed time until subjects were woken up for performing the BR measures after 2 hr of sleep, whereas during the MDmorn session, EEG was acquired in the 2 hr of dark exposure. According to EEG recordings, in the night sessions participants fell asleep easily and fast (sleep latency 9±2 min – mean ± SE), showed a normal organization of sleep structure (37±3% spent in N2 stage, and 50±3% spent in N3 stage, on average), exhibited none (8 out of 15 subjects in the MDnight and 4 out of 15 subjects in the Cnight) or few minutes of REM sleep (9±2 min) and none of them had wakefulness episodes after sleep onset.

For the MDmorn session, the subject was welcomed in the laboratory around 9 AM, shortly thereafter he/she was tested for the first BR measurement and then subjected to MD for 2 hr, during which he/she could read or watch movies. Consequently around 11.30 the patch was removed, the volunteer was tested for a second BR measurement immediately followed by 2 hr of resting by lying in bed in the dark. After 2 hr (1.30 PM), the BR was again measured multiple times according to a sequence suitable for reconstructing the extinction curve of the MD effect (the sequence lasted 30 min). In the MDmorn session, none of the subjects were allowed to feel asleep as EEG was

monitored in real time and, in case of drowsiness signs (EEG slowing to theta), a bell tolled in the resting room. Having done this session in the morning, the use of the bell was extremely occasional: some subjects never showed signs of falling asleep, others were alerted at most three times with the bell in the last part of the resting state. A diagram of the experimental paradigm for the MDnight is reported in *Figure 1A*.

## MDnight and Cnight EEG processing

To prepare EEG signals for the planned analysis, they underwent some pre-processing. EEG pre-processing and analyses were performed using tailored codes written in Matlab (MathWorks, Natick, MA); scalp maps were obtained using EEGLAB Toolbox functions (*Delorme and Makeig, 2004*).

For night sessions, scalp EEG signals were re-referenced to the average mastoid and scored according to the AASM criteria (*Marshall et al., 2006*). In particular, artefacts related to movements or muscle twitches were detected during the signal inspection performed for visual sleep stage scoring. EEG segments affected by artefacts were thus excluded from the analysis. Moreover, temporary or permanent decline in individual channel signal quality (often due to instability or loss of contact with the scalp during recordings) was studied on the basis of the signal statistical moments (EEGLAB Toolbox *Censor et al., 2006*). Channels characterized by outliers in the statistical moments were classified as 'bad' channels and excluded from analysis. At the end of these pre-processing steps, all recordings have shown less than 10% of artefact-contaminated segments.

Sleep macrostructure was evaluated by extracting a set of time-domain parameters from the sleep staging annotations: sleep latency (time length of the transition from lights-off to the first N2 sleep episode, min); wake episodes after sleep onset (WASO duration, min); shift phase (sleep stage shift per time unit), sleep fragmentation (awakenings and shifts to lighter sleep stages per time unit), N2, N3 and REM (rapid eye movement) stages duration (min); and REM latency (time from sleep onset to the first REM sleep episode, min). Moreover, according to the sleep stage scoring, artefact-free NREM (N2 and N3) segments were analysed for estimating power band content and identifying sleep patterns such as SSOs and sleep spindles.

For the EEG signals analysis, only segments classified as NREM (N2 and N3) and free of artefacts were considered. For power band content of NREM sleep, two frequency bands of interest were considered: SWA (0.5–4 Hz) and sigma (σ: 9–15 Hz). Power densities were estimated applying a Hamming-windowed FFT on 10 s consecutive EEG segments and log-transformed (dB). For each segment, electrode and band, the absolute power was estimated by averaging over its frequency bins. The absolute power of each band and electrode was then obtained by averaging among segments.

## MDmorn EEG processing

For the MDmorn session, EEG was pre-processed for managing diffuse artefacts with reconstruction of virtually clean traces. Thus, EEG signals were high-pass filtered at 0.1 Hz (Chebyshev II filter) and notch filtered at 50 Hz and its first harmonic (100 Hz). Channels located on the forehead and cheeks which mostly contribute to movement-related noise were discarded, thus retaining 107 channels out of 128 (*Piarulli et al., 2010*). Epochs with signals exceeding 100 µV were automatically discarded; retained signals were visually inspected for the removal of artefacts and noisy channels. Rejected signals were substituted with signals obtained via spline interpolation (*Junghöfer et al., 2000*) and further submitted to the Independent Component Analysis for separating and removing components expressing eye movements, heart beats, line, and channel noise – component selection also supported by the AI system of ICLabel (*Pion-Tonachini et al., 2019*). After artefact removal procedures, EEG signals were re-referenced to the average of the mastoid potentials (*Piarulli et al., 2010*; *Laurino et al., 2014*).

The artifact-free EEG segments analysed for estimating power band content in five frequency bands of interest were considered: theta (4–8 Hz), alpha (8–12 Hz), low beta (12–20 Hz), high beta (20–30 Hz), and gamma (30–45 Hz). Power densities were estimated by applying a Hamming-windowed FFT on 4 s consecutive EEG segments and log-transformed (dB). For each segment, electrode and band, the absolute power was estimated by averaging over its frequency bins. The absolute power of each band and electrode was then obtained by averaging among segments.

## SSO detection and characterization

SSO events within NREM sleep periods were detected and characterized using a previously published and validated algorithm (*Piarulli et al., 2010*; *Menicucci et al., 2009*). In summary, the algorithm first identifies full-fledged SSOs: each wave should comprise (1) two zero crossings separated by 0.3–1.0 s, the first one having a negative slope; (2) a negative peak between the two zero crossings with a voltage less than −80 μV; (3) a negative-to-positive peak amplitude of at least 140 μV. An SSO event is defined when simultaneous SSOs (tolerance of up to 200 ms delay between negative wave peaks) are recognized on multiple channels Then, detected SSO events are completed by clustering full-fledged SSOs with concurrent similar waves, even if sub-threshold detected on the other EEG channels. These detection criteria naturally include all K-complexes (*Massimini et al., 2004*) but can also complete detection with otherwise neglected small waves.

For each subject, night session and electrode channel, SSOs were characterized by the following parameters: detection rate (number of waves per minute), negative-to-positive peak amplitude (NP amp), slope from the negative peak (slope+) (*Menicucci et al., 2013*). Moreover, the sigma activity (9–15 Hz) expressed in the 1 s window preceding each SSO was estimated as a known thalamocortical entrainment marker functioning as precursor of SSOs emergence (*Menicucci et al., 2015*).

## Sleep spindle detection and characterization

The sleep spindle recognition was carried out according to the approach proposed and validated by *Ferrarelli et al., 2007*, with some minor adaptations; the actual procedure is summarized below.

EEG data for all NREM sleep periods were band-pass filtered between 12 and 16 Hz (−20 dB at 11 and 17 Hz) and for each channel, the upper and lower envelopes of the filtered signal were derived. The spindle detection was based on the signal derived as the point-by-point distance between the upper and the lower envelopes (signal amplitude). Because signal amplitude varies between channels, for each NREM period and channel a value was estimated as the mean signal amplitude increased by twice its standard deviation. Thus, for each channel a threshold was defined as the weighted average over the values calculated for each period, with the weights corresponding to the period lengths. Sleep spindles were thus identified as the fluctuations in the amplitude signal exceeding two times the threshold. Based on the detected spindles, for each EEG channel, the measures characterizing each sleep recording were the density (spindle events per time unit) and the spindle power. The spindle power was derived as the average over the spindle events of the sigma band (12–16 Hz) power of each detected wave.

## Binocular rivalry

The perceptual reports recorded through the computer keyboard were analysed using Matlab, the mean phase duration and the total time of perceptual dominance of the visual stimuli presented to each eye and mixed percepts were computed for each participant and each experimental block. The 3-min blocks acquired after the morning awakening were binned as follows: 0–8, 10–18, 30 min.

The effect of MD was quantified by computing a DI, The effect of MD was quantified by computing a DI, using the same formula in *Lunghi et al., 2015b*, which demonstrated that DI correlates with the change in GABA concentration in the visual cortex of adult participants. The DI summarizes in one number the change in the ratio between deprived and non-deprived eye mean phase duration following MD relative to baseline measurements according to the following equation:

$$\text{Deprivation Index (DI)} = \left(baseMPD_{dep-eye}/depMPD_{dep-eye}\right) * \left(depMPD_{Ndep-eye}/baseMPPD_{Ndep-eye}\right) \quad (1)$$

In *Equation 1*, MPD is the mean phase duration computed in seconds, base stands for baseline measurements, dep for measurements acquired after MD. A DI value equal to 0 represents no change in the ratio between dominant and non-dominant eye mean phase duration, while a value >1 represents a decrease in dominant eye predominance and a value <1 an increase in dominant eye predominance during BR.

## Control experiment

In order to investigate the influence of the circadian rhythm on visual homeostatic plasticity, the effect of MD was tested for each participant in two different days, once in the morning and once in the

evening (the order of the conditions was counterbalanced across participants). In the morning session, the 2 hr MD started at 9 AM, while in the evening session, MD started at 8 PM. Each deprivation session was preceded by 2×3 min baseline BR blocks. After patch removal, we measured BR continuously for 18 min in four separate 180 s blocks, giving a short break every 2 min. Three-minute blocks of rivalry were tested again 30, 45, 60 90, and 120 min after restoration of normal binocular sight. The 3-min blocks acquired after MD were binned as follows: 0–8, 10–18, 30–48, 60–93, and 120–123 min. Each participant therefore spent one morning (from 8.30 AM to 1 PM) and one evening (from 7.30 PM to 0.00 AM) in the lab. All the procedures and analyses were the same as the main study.

## Statistical analyses

The variation of DIs measured during the two-night experiments was tested using repeated-measures ANOVA with one factor (TIME) with five levels (*before, after, morning1, morning2, morning3*). In order to compare directly the decay of the deprivation effect in the MDnight and in the MDmorn condition, we used a repeated-measures ANOVA with two factors: TIME (with two levels: before and after) and CONDITION (MDnight and MDmorn). For the control experiment, we used a repeated-measures ANOVA with two factors: TIME (with five levels corresponding to the five measurements obtained after deprivation, i.e, before, after, morning1, morning2, morning3) and CONDITION (MDnight and MDmorn). One-sample two-tailed t-tests tests were used for post hoc tests, against the H0 mean = 1. The *Benjamini and Hochberg, 1995*, correction for multiple comparisons was applied for post hoc tests, effect size was estimated computing Cohen's d.

The variation of sleep characteristics according to the MD intervention and the dependence on the one hand on the acquired eye dominance (as measured by the DI before sleep) and on the other on the residual dominance evaluated after 2 hr of sleep (as measured by the DI after sleep) was evaluated considering three (ROIs) regions of interest: occipital ROI, prefrontal ROI, and sensory-motor control ROI. These ROIs were selected on the literature evidence that MD alters activity over extensive occipito-parietal network of visual areas. The ROI in the prefrontal cortex was selected given its key role in organizing the ripple-mediated information transfer from hippocampus during NREM sleep (*Helfrich et al., 1995*). ROI-based analysis is not only in line with the assumptions of this work but also meets the low spatial resolution of EEG. Furthermore, employing ROIs allowed us to decrease the number of statistical tests (for each EEG feature, we have evaluated data on three ROIs instead of 90 electrodes). Finally, it made possible to manage inter-individual variability of anatomical brain structures. In particular, the large anatomical variability of V1 orientation implies a high variability in the dipole orientation and the related voltage over electrodes from Oz to CPz. The electrodes in the HydroCel Geodesic Sensor Net belonging to each ROI are shown in *Figure 3—figure supplement 1*.

For the effect of MD intervention on cortical activity during NREM sleep, the Wilcoxon signed rank test was applied on all the EEG sleep features. To this aim, each sleep feature was averaged over the electrodes belonging to occipital and sensorimotor control ROIs and thus compared between nights. Only variations between conditions whose statistical significance survived the false discovery rate (FDR *Benjamini and Hochberg, 1995*) correction for multiple testing were considered.

For the dependence on the individual plasticity induced by MD, the non-parametric Spearman's correlation was determined between DI before sleep, and each EEG sleep feature averaged over the occipital and sensorimotor control ROIs. Only correlations whose statistical significance survived the FDR correction for multiple testing were considered. Analogously, the association with the individual residual eye dominance after sleep was determined by Spearman's correlation calculated between DI after and each EEG sleep feature averaged over the occipital, the prefrontal, and sensorimotor control ROIs. Also for this group of tests, only correlations whose statistical significance survived the FDR88 correction were considered.

For all tests, the level of statistical significance after FDR correction was set at p<0.05; and the FDR was set equal to p=0.05.

## Acknowledgements

The research presented in this study was funded by:

European Research Council (FPT/2007–2013) under grant agreements 338866 'Ecsplain' and 832813 'GenPercept.', from the European Research Council (ERC) under the European Union's Horizon 2020 research and innovation programme (No 948366 – HOPLA), PRIN 2015 from MIUR, and the French

National Research Agency (ANR: AAPG 2019 JCJC, grant agreement ANR-19-CE28-0008, PlaStiC, and FrontCog grant ANR-17-EURE-0017).

## Additional information

### Funding

| Funder | Grant reference number | Author |
|---|---|---|
| FP7 | 338866 - Ecsplain | Maria Concetta Morrone |
| ERC | 948366 - HOPLA | Claudia Lunghi |
| FP7 | 832813 - GenPercept | Maria Concetta Morrone |
| French National Research Agency | ANR-19-CE28-008, PlaStiC | Claudia Lunghi |
| MIUR | PRIN 2015 | Maria Concetta Morrone |
| French National Research Agency | FrontCog ANR-17-EURE-0017 | Claudia Lunghi |

The funders had no role in study design, data collection and interpretation, or the decision to submit the work for publication.

### Author contributions

Danilo Menicucci, Conceptualization, Data curation, Formal analysis, Investigation, Methodology, Writing - original draft, Writing - review and editing; Claudia Lunghi, Conceptualization, Data curation, Formal analysis, Funding acquisition, Investigation, Methodology, Writing - original draft, Writing - review and editing; Andrea Zaccaro, Data curation, Investigation, Writing - review and editing; Maria Concetta Morrone, Conceptualization, Supervision, Funding acquisition, Methodology, Writing - original draft, Writing - review and editing; Angelo Gemignani, Conceptualization, Formal analysis, Supervision, Methodology, Writing - original draft, Writing - review and editing

### Author ORCIDs

Danilo Menicucci (iD) http://orcid.org/0000-0002-5521-4108
Claudia Lunghi (iD) http://orcid.org/0000-0003-3811-5404
Andrea Zaccaro (iD) http://orcid.org/0000-0002-0409-7132
Maria Concetta Morrone (iD) http://orcid.org/0000-0002-1025-0316

### Ethics

All eligible volunteers signed an informed written consent. The study was approved by the Local Ethical Committee (Comitato Etico Pediatrico Regionale-Azienda Ospedaliero-Universitaria Meyer-Firenze), under the protocol "Plasticità del sistema visivo" (3/2011) and complied the tenets of the Declaration of Helsinki.

### Decision letter and Author response

Decision letter https://doi.org/10.7554/eLife.70633.sa1
Author response https://doi.org/10.7554/eLife.70633.sa2

## Additional files

### Supplementary files

• Supplementary file 1. Table: Sleep macrostructural parameters: monocular deprivation night (MDnight) and control night (Cnight) descriptive statistics.

• Supplementary file 2. Table: Features comparison between conditions and correlations with plasticity indices for the relevant ROIs.

• Transparent reporting form

• Source code 1. Matlab files with data and codes for statistics and final results.

### Data availability

All data included in manuscript and supporting files.

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
