## [Editor Report]

Menicucci and colleagues investigated the potential role of sleep in the homeostatic plasticity of ocular dominance in adult humans. This is a careful study that should be of broad interest to those studying adult cortical plasticity, particularly in vision. The study shows that sleep can maintain the changes in ocular dominance obtained after applying an eye patch on the dominant eye for two hours, which contrasts with the rapid decline of these changes during quiet wake in darkness. The authors further report correlations between sleep oscillations and the magnitude of the plasticity effect. Overall, these results implicate sleep in a new form of plasticity.

---

## [Decision Letter]

**Decision letter after peer review:**

Thank you for submitting your article "Sleep Slow Oscillations and Spindles encode ocular dominance plasticity and promote its consolidation in adult humans" for consideration by *eLife*. Your article has been reviewed by 3 peer reviewers, one of whom is a member of our Board of Reviewers, and the evaluation has been overseen by Chris Baker as the Senior Editor. The reviewers have opted to remain anonymous.

Essential revisions:

1) While the authors argue that NREM sleep consolidates the effect of MD, the outcomes of the study suggest that this would not be considered consolidation by most definitions. Since the effect is gone in 6 hours or so, it doesn't seem that there is any long-term effect of MD – quite different from sleep-dependent consolidation. There may be a slowing of the rate of decay (this would need to be statistically tested by comparing sleep and awake controls), but the effects of MD lessen over time in all cases. Although the findings of the effects of sleep on ocular dominance plasticity are interesting, the terminology used – e.g. consolidation, homeostatic vs. Hebbian – do not seem well founded based on data.

2) Experimental conditions do not seem to be sufficient to show the effect of sleep itself. To investigate the role of sleep in MD, the authors compared the deprivation index between two sessions, the main MDN and a control MDM session. The experimental designs for these two sessions were quite different; MD was conducted in the evening in MDN, whereas it was conducted in the morning in MDM. Since there may be circadian effects, the comparisons between these sessions are not sufficient in investigating the effect of sleep itself (it could be merely due to circadian effect). There needs to be a wake-control group where MD and binocular rivalry sessions are performed at the same times of day as in MDN session to investigate whether the results (Figure 1B1, B2) are specific to sleep or not.

3) Important statistical tests (and information on them) to investigate the effect of sleep are missing. For the MD data in Figure 1, there seems to be a continuous change from the "Before" timepoint over time, and this is not seen after dark exposure spent awake (b3). For example, are the "after 2h" time points in B1 and B3 statistically different from one another? Was there an interaction effect? In addition, the DI in the control night session seem to be larger than "1". While correlation coefficients were measured (Figure 2), there are no direct tests of changes across conditions (MD vs. control). Thus, it is unclear how sleep oscillations were changed by MD. Did the strength of sleep spindles or slow oscillations changed by MD? To correct for multiple comparisons, the authors used the FDR method. But it is unclear how this was applied. How were the multiple tests defined? Since the number of participants is small, the authors could indicate effect sizes.

4) How the selection of regions of interest was performed is unclear. The authors selected particular electrodes to explore the relationship between MD-related plastic changes and sleep features. How were these electrodes selected? Furthermore, EEG activities were averaged across hemispheres in the occipital region. Previous studies (e.g. Lunghi et al., 2011; Binda et al., 2018) have demonstrated that MD is associated with up-scaling of the deprived eye and with down-scaling of the non-deprived eye. I wonder whether sleep slow oscillations and / or spindles are modulated locally in the deprived occipital region? To answer the first question raised by the authors (how MD affects subsequent sleep), wouldn't it be important to compare between deprived vs. non-deprived regions? It is recommended that the authors look at the effect in more depth at the electrodes level. In addition, what do the authors make of figure 2 B4, given that it is frontal rather than occipital and correlated with DI after rather than before? what does this relationship mean?

5) Sleep architecture needs to be provided, as it would be helpful for seeing relationship (or lack thereof) between these measures and prior REM or NREM time. Furthermore, information on some important methods is missing, for example, there was no information as to how EOG and EMG were measured. These measurements could be crucial for detecting REM sleep in humans.

*Reviewer #1 (Recommendations for the authors):*

– The authors could investigate the relationships between region-specific brain activities and the deprivation index.

– There needs to be a wake-control group where MD and binocular rivalry sessions are performed at the same times of day as in MDN session to investigate whether the results (Figure 1B1, B2) are specific to sleep or not.

– I would recommend that the term consolidation and suggested neural mechanisms be clarified or revised. Would it be possible that homeostatic processing is prolonged or the decay rate is delayed by sleep?

– The authors previously showed that MD decreased GABA in the visual cortex (Lunghi et al., 2015). It may be interesting to discuss whether some changes in GABA level during sleep worked to enhance GABA-related processing.

– High density EEG recordings were monitored (page 17). Please describe how EOG and EMG were measured in the experiment, since these measurements could be crucial for detecting REM sleep.

– Please show sleep variables (e.g. the durations for each sleep stage).

– Please indicate the experimental design for the other control sessions (CN and MDM) in a figure. I think it would be helpful for the readers if all the conditions are shown in one figure (e.g. in Figure 1A).

– Please provide information as to approximately what time each phase (MD, binocular rivalry) occurred in MDN, CN, and MDM sessions (page 18).

– Page 18, 9AM to 14 AM at the laboratory. Is this 14PM?

*Reviewer #2 (Recommendations for the authors):*

My specific concerns and recommendations are as follows:

1) Figure 1, B1-3 – can indicators of statistical test results be put into the figure, for clarity?

2) For control night data in Figure 1 B1-2 – Are some of the control timepoint values statistically different from 1? It looks like there might be a deviation that either reflects (more likely) circadian time, or otherwise reflects an effect of wake time?

3) For the MD data – since there seems to be a continuous change from the "Before" timepoint over time, and this is not seen after dark exposure spent awake (b3), is this definitely an effect of sleep, or could it also be an effect of time? e.g. are the "after 2h" time points in B1 and B3 statistically different from one another? A related question is since the deprivation index is lower in B3, is there a statistically significant effect of sleep, rather than time at all. Are the "before" timepoints in B1 and B3 statistically different?

4) Is there quantification of the sleep architecture in both B1 and B3? This should be provided, as it would be helpful for seeing relationship (or lack thereof) between these measures and prior REM or NREM time.

5) A general comment on the measure – I don't think this would be considered "consolidation" by most definitions. Consolidation usually means retention, or enhancement of plastic changes initiated during learning/experience. There may be a slowing of the rate of decay (this would need to be statistically tested by comparing sleep and awake controls), but the effects of MD lessen over time in all cases. It doesn't seem that there is any long-term effect of MD – quite different from sleep-dependent consolidation that was reported in critical period cat and adult mouse V1.

6) What do the authors make of figure 2 B4, given that it is frontal rather than occipital and correlated with DI after rather than before? what does this relationship mean?

7) For Figure 2 – this is definitely showing that there is a relationship between prior plasticity in wake, and subsequent NREM sleep parameters, but is anything related to the change in this index ACROSS sleep? Such a change would be a convincing indication that there is indeed any kind of plasticity (or even slowing of decay or prior changes) occurring in the circuit during sleep.

8) On page 10: "..the power was reduced in participants showing a lower response to MD. This result, together with the other σ power estimates, reinforces the idea behind a modulation of thalamocortical interaction as a homeostatic reaction to MD. " A more parsimonious interpretation (in light of the lack of additional change during sleep – over and above what is seen in MD alone) is that propagation of activity through the circuit changes if the wiring has previously been changed. So this statement is not really an obvious interpretation.

*Reviewer #3 (Recommendations for the authors):*

1. As mentioned above, it would be good to clarify the hypothesis motivating this study (p. 4, e.g. "we expected to observe a relation between SSO, σ oscillations and plasticity in visual area") and the putative mechanisms the authors derived from their results (exposed in the Discussion). I do think linking this result with previous findings on active memory consolidation, involving hippocampal ripples, sleep spindles and cortical slow waves is interesting. However, I struggle to see how the same mechanisms could be involved in MD-related plasticity. From the literature presented by the authors, it seems this plasticity affects the primary visual cortex, so not the form of hippocampus-dependent memory typically benefiting from active consolidation. In addition, I do not see what would be replayed here and how sleep spindles, given what we know about their involvement in memory consolidation, would act to consolidate this form of plasticity.

2. The authors present their results as a "consolidation" but I do not see what is the argument for consolidation. To me, it seems that what they show is a maintenance of the effect during sleep. But they do not show a modification of the memory trace (enhancement, transfer, abstraction) that is usually being referred to as "consolidation".

3. Regarding the MD-related plastic change, I am wondering if the authors could expand on the direction of the effect found in the different conditions. In particular, in the control night session, why would they obtain the reverse effect (DI>1)? Since there is no MD done in the control session, should the DI be equal to 1?

4. The authors used ROI to explore the relationship between MD-related plastic changes and sleep features. How were these ROI defined? Why using ROI and not analyses at the electrode level? The topographical maps of the correlation coefficients seem to show clusters of electrodes in occipital area so an analysis at the electrode level should provide similar results but would rely on less arbitrary settings.

5. To correct multiple comparisons, the authors used the FDR method. But it is unclear how this was applied. How were groups of multiple tests defined? If the number of tests is low, the FDR is not, I think, the best approach to correct for multiple comparisons. It is typically used when correcting for hundreds or thousands of tests (e.g. in fMRI).

---

## [Author Response]

Essential revisions:1) While the authors argue that NREM sleep consolidates the effect of MD, the outcomes of the study suggest that this would not be considered consolidation by most definitions. Since the effect is gone in 6 hours or so, it doesn't seem that there is any long-term effect of MD – quite different from sleep-dependent consolidation. There may be a slowing of the rate of decay (this would need to be statistically tested by comparing sleep and awake controls), but the effects of MD lessen over time in all cases. Although the findings of the effects of sleep on ocular dominance plasticity are interesting, the terminology used – e.g. consolidation, homeostatic vs. Hebbian – do not seem well founded based on data.

We thank the reviewer for raising this issue. We agree that the data show a substantial slowing of the decay process of the MD effects after the removal of the patch. The present data indicate that specifically the sleep condition and not merely darkness is responsible for the maintenance of the MD-induced effect during the night. Therefore, we gladly adhere to the request and propose to say that sleep stabilizes/maintains the effect of MD as long as sleep itself persists. Accordingly, we made the resulting changes throughout the MS, starting with the new title.

About homeostatic vs Hebbian plasticity, there is a quite large agreement in the literature stating that indeed the effects are different. Now we make clear in the text that Hebbian plasticity is usually associated to the boost of most successful signals in driving a neuronal response or a behavior. Here the MD produced a boost of the unused, and probably silent, eye and as such the boost it is very difficult to explain in term of Hebbian plasticity. We make now this clear in the introduction.

2) Experimental conditions do not seem to be sufficient to show the effect of sleep itself. To investigate the role of sleep in MD, the authors compared the deprivation index between two sessions, the main MDN and a control MDM session. The experimental designs for these two sessions were quite different; MD was conducted in the evening in MDN, whereas it was conducted in the morning in MDM. Since there may be circadian effects, the comparisons between these sessions are not sufficient in investigating the effect of sleep itself (it could be merely due to circadian effect). There needs to be a wake-control group where MD and binocular rivalry sessions are performed at the same times of day as in MDN session to investigate whether the results (Figure 1B1, B2) are specific to sleep or not.

We thank the reviewer for raising this important point. We performed the dark exposure experiment in the morning because we wanted to minimize the occurrence of sleep during the two hours spent by participants lying down in complete darkness. Preventing sleep under these conditions in the late evening would have been extremely challenging.

In order to investigate a possible influence of the circadian rhythm on visual homeostatic plasticity and its decay over time, we have performed an additional experiment. In this experiment, we have tested the effect of 2h of monocular deprivation in the same group of participants either early in the morning or late at night, at times comparable to the MDnight (previously indicated as MDN) and MDmorn (previously indicated as MDM) conditions in the main study. We report the results of this control experiment in the supplementary materials. We found that the effect of monocular deprivation follows a similar time course for the two conditions (ocular dominance returns to baseline levels within 120 minutes after eye-patch removal). Moreover, we also report that the effect of MD is slightly (but significantly) larger in the morning, compared to the evening. The results of this experiment rules out a contribution of circadian effects and reinforces the evidence of a specific effect of sleep in maintaining visual homeostatic plasticity.

3) Important statistical tests (and information on them) to investigate the effect of sleep are missing. For the MD data in Figure 1, there seems to be a continuous change from the "Before" timepoint over time, and this is not seen after dark exposure spent awake (b3). For example, are the "after 2h" time points in B1 and B3 statistically different from one another? Was there an interaction effect?

Thank you for this important suggestion. We now directly compare the effect of monocular deprivation and its decay after two hours in the sleep vs dark exposure condition (MDnight vs MDmorn). In order to improve the clarity of the results we now plot the results of the two conditions in the same graph (Figure 2). We found a significant interaction effect between the factors TIME (before and after) and CONDITION (MDnight and MDmorn), indicating a specific role of sleep in stabilizing the decay of short-term monocular deprivation.

We also clarify in the methods and in the Results sections the statistical analysis that we used.

In addition, the DI in the control night session seem to be larger than "1".

In the control night session, overall, the DI does not significantly differ across the different times. Because of the lack of a main effect, we did not think it necessary to perform post-hoc tests. However, on the reviewers’ input, we performed some exploratory post-hoc analyses and found that the DI measured in the control night before sleep was significantly larger than 1. We now report this result in the manuscript. We believe that this transient change in ocular dominance might reflect either a circadian effect or a transient form of adaptation due to the repetition of the binocular rivalry test, as reported in Klink et al., (2010).

While correlation coefficients were measured (Figure 2), there are no direct tests of changes across conditions (MD vs. control). Thus, it is unclear how sleep oscillations were changed by MD. Did the strength of sleep spindles or slow oscillations changed by MD?

We did not identify any main effect of MD on sleep characteristics (Wilcoxon test with FDR correction as indicated in Materials and methods, section Statistical analyses). The revised version of the Results section now explicitly states that we have no main effects; moreover, the absence of main effects on sleep is discussed in the middle of the section "Sleep and the plastic potential of visual cortex" of the Discussion.

To correct for multiple comparisons, the authors used the FDR method. But it is unclear how this was applied. How were the multiple tests defined?

We thank the reviewer for highlighting this unclear point. In the revised version of the Statistical analyses section, we have provided missing details of the procedure used for handling false positives due to multiple testing. We applied the FDR correction for each test that we performed. For example, “at which time points does dominance remain significantly different from baseline?” or, “which EEG feature and in which area of the scalp shows changes significantly dependent on plasticity induced by monocular deprivation?” For each of these analyses, we made a group of tests to which Benjamini and Hochberg's FDR correction was then applied (for the first example, the correction is based on the number of points at which ocular dominance was assessed until the morning; for the second example, the number of EEG features examined multiplied by the number of areas in which they were assessed)

Since the number of participants is small, the authors could indicate effect sizes.

Thank you. We added effect sizes for all statistical tests in the main text. The revised version of the Statistical analyses section now mentions the calculation of the effect size (Cohen’s d for post-hoc t-tests and eta squared for ANOVAs).

4) How the selection of regions of interest was performed is unclear. The authors selected particular electrodes to explore the relationship between MD-related plastic changes and sleep features. How were these electrodes selected?

The selection of regions of interest (ROIs) stems from the hypotheses of the work. Given low spatial resolution of the EEG (volume conduction effect) and having literature evidence that MD alters activity over extensive bilateral visual areas, we hypothesized that MD may alter cortical activity in occipito-parietal areas (and not at specific electrodes) during sleep. Also, another ROI was identified in the prefrontal cortex for its key role in organizing the ripple-mediated information transfer from hippocampus during NREM sleep.

ROI-based analysis also allowed us to decrease the number of statistical tests (we evaluate 3 ROIs instead of 90 electrodes) and made it possible to manage inter-individual variability of brain structures, in particular the large anatomical variability of V1 orientation implying a variably oriented dipole and a variable maximal representation of visual potentials over electrodes from Oz to CPz. In the Statistical analyses section of Materials and methods we have provided reasons to support this choice.

From the spatial distribution of single electrodes r_s_ coefficients (see new Figure 3 and 4) we see a good homogeneity of the values within the ROIs: this, a posteriori, validates the choice of the ROIs used for the analysis. In the revised version of the figure caption we have emphasized that scalp maps depict single electrode correlation distribution. In addition, we have included supplementary figures, which shows boxplots for the electrodes inside each ROI to evaluate the coherence between electrodes of correlations.

Furthermore, EEG activities were averaged across hemispheres in the occipital region. Previous studies (e.g. Lunghi et al., 2011; Binda et al., 2018) have demonstrated that MD is associated with up-scaling of the deprived eye and with down-scaling of the non-deprived eye. I wonder whether sleep slow oscillations and / or spindles are modulated locally in the deprived occipital region? To answer the first question raised by the authors (how MD affects subsequent sleep), wouldn't it be important to compare between deprived vs. non-deprived regions?

In humans, large scale visual cortical organization is based on the visual field map and the monocular recipient region of the various cortices is very small and relegated to the far visual periphery: neurons whose visual receptive fields lie next to one another in visual space are located next to one another in cortex, forming one complete representation of contralateral visual space, independently of the eye from which the visual information comes. However, at finer scales ocular dominance columns exist and Binda et al., (2018) showed that in adult humans MD boosts the BOLD response to the deprived eye, changing ocular dominance of V1 vertices, consistent with homeostatic plasticity. So there is not a single area in the human cortex that can receive input from ONLY one eye that can be measured by EEG or EMG.

It is recommended that the authors look at the effect in more depth at the electrodes level.

The need for ROIs is based on the interindividual variability of brain structures, in particular the large anatomical variability of V1 orientation implying a variably oriented dipole and a variable maximal representation of visual potentials over electrodes from Oz to CPz.

Also, a ROI would summarize a MD cortical effect expected to extend bilaterally over the occipital lobe (present in V1, V2, V3 and V4). Finally, the ROIs allowed to cope with the volume conduction effect that strongly limits EEG spatial resolution.

With these limitations in mind, we very gladly adhere to the reviewer's request to evaluate the effects on individual electrodes in more detail.

Figure 2 in the previous version of the manuscript (now figures 3 and 4) showed the correlation values (r_s_) calculated for each single electrode. From the spatial distribution of single electrodes r_s_ coefficients we see a good homogeneity of the values within the ROIs: this, a posteriori, validated the ROIs selection. In the revised version of the figure caption we have emphasized that scalp maps depict single electrode correlation distribution. In addition, for the revised manuscript, we have included supplementary figures, which shows boxplots for the electrodes inside each ROI to evaluate the coherence between electrodes of correlations.

In addition, what do the authors make of figure 2 B4, given that it is frontal rather than occipital and correlated with DI after rather than before? what does this relationship mean?

In the present work, after having demonstrated that sleep specifically maintains the ocular dominance induced by short term MD, two different but intermingled issues have arisen: (1) how MD affects subsequent sleep, and (2) how sleep contributes to maintenance of the acquired ocular dominance. Given the supporting literature, for (1) we hypothesized that we could find changes in the visual areas (posterior ROI) while for (2) we searched for an association in the prefrontal areas due to the dense connections with the hippocampus. We also hypothesized that the effect of MD on sleep should be proportional to the induced eye dominance and then for (1) we correlated DI before with the features of the sleep EEG, while for (2) we try to explain the residual effect of MD after 2 hours and thus we consider the DI after. Figure 4 shows the only significant association we found between residual dominance after 2 hours (DI after) and a sleep feature (spindle density).

On the basis of the reviewer's remark, we decided to present two separate figures (figure 3 and 4): one specifically for associations with DI before, the other for those with DI after. In each new figure we include the EEG features that resulted significant associated with at least one among DI before and after.

5) Sleep architecture needs to be provided, as it would be helpful for seeing relationship (or lack thereof) between these measures and prior REM or NREM time.

We have provided a summary table of sleep architecture in the revised version of the Supplementary Materials. The table shows descriptive statistics of sleep architecture for the MDnight and Cnight conditions. Also, we report the result of the paired comparison between the MD and the Control night and the Spearman correlations between the deprivation indices (DI before and DI after) and the changes between the nights in sleep architecture. Tests indicate that MD does not produce any main effect on the sleep architecture and that there are no substantial associations found between sleep architecture parameters and deprivation indices. Thus, it appears that changes in SSO and spindle frequency and amplitude did not lead to an alteration in the amount of N2 or N3 sleep, as we might expect. At the beginning of the Results section we now refer to the table and to the lack of statistically significant effects.

Furthermore, information on some important methods is missing, for example, there was no information as to how EOG and EMG were measured. These measurements could be crucial for detecting REM sleep in humans.

We apologize for this imprecision, the information has now been added in High density EEG recordings of Material and Methods section. The EEG system employs a high spatial density electrode system with full head coverage including periocular, cheekbone and neck sensors. This allows the detection of both vertical and horizontal eye movements and muscle tone (both from the zygomaticus major muscles and upper trapezius muscles on the neck) directly from the EEG cap.

Reviewer #1 (Recommendations for the authors):– The authors could investigate the relationships between region-specific brain activities and the deprivation index.

The electrophysiological study of region-specific MD effects would be very intriguing. However, some limitations need to be taken into account: (1) in humans, neurons whose visual receptive fields lie next to one another in visual space are located next to one another in cortex, forming one complete representation of contralateral visual space, independently of the eye from which the visual information comes. (2) associations between DI and activity changes in individual electrodes distributed over the whole scalp were studied and presented in the maps in Figure 2. Unfortunately, taking into account the 90 electrodes, single electrode correlations were not significant after multiple testing corrections (Statistical non-Parametric Mapping corrections) even for rho~0.7.

– There needs to be a wake-control group where MD and binocular rivalry sessions are performed at the same times of day as in MDN session to investigate whether the results (Figure 1B1, B2) are specific to sleep or not.

We have performed a control wake experiment in which we tested the effect of short-term monocular deprivation in different times of the day to investigate the specificity of sleep. We report the results of this experiment in the main text and in supplementary materials.

– I would recommend that the term consolidation and suggested neural mechanisms be clarified or revised. Would it be possible that homeostatic processing is prolonged or the decay rate is delayed by sleep?

We thank the reviewer for raising this issue. We agree that the data just show a substantial delay in the decay process of the MD effects after the removal of the patch. The present data indicate that specifically the sleep condition and not merely darkness would be responsible for the maintenance of the MD-induced effect during the night. Therefore, we gladly adhere to the request and propose to say that sleep stabilizes/maintains the effects of MD as long as sleep itself persists. Having said that, we would like to point out that the MD boost in amblyopic patients gets consolidated for up to one over a period of a year and increases across night sleep as we reported in Lunghi, Sframeli et al., (2019). We believe that consolidation may be the real phenomenon, but we agree with the reviewer that our data did not directly address this question.

– The authors previously showed that MD decreased GABA in the visual cortex (Lunghi et al., 2015). It may be interesting to discuss whether some changes in GABA level during sleep worked to enhance GABA-related processing.

Thank you for an important suggestion. We have added the following paragraph to the discussion session:

“Both homeostatic plasticity^41,69^ and SSO^42,70^ are linked to GABAergic inhibition, the observed effect could therefore in principle be mediated by a change in excitation/inhibition balance in the visual system. Importantly, in humans, GABA concentration measured after 2 hours of MD in V1 decreased proportionally to the observed ocular dominance shift^30^, and slow wave activity expression increases in response to GABA agonists administration^71^. Interestingly, a recent study^72^ reported that complementary changes in the excitation/inhibition balance measured in the visual cortex of adult humans during NREM and REM sleep are correlated with visual plasticity induced by perceptual learning, further pointing to a leading role of GABAergic inhibition in mediating these two phenomena. We speculate that the alteration of SSO shape and density observed at the occipital level during the sleep immediately following monocular deprivation might reflect the change in GABA concentration induced by deprivation and be a neurophysiological marker of the interindividual variability in the level of plasticity in adult humans.”

– High density EEG recordings were monitored (page 17). Please describe how EOG and EMG were measured in the experiment, since these measurements could be crucial for detecting REM sleep.

Missing information has now been added in High density EEG recordings of Material and Methods section. The EEG system employs a high spatial density electrode system with full head coverage including periocular, cheekbone and neck sensors. This allows the detection of both vertical and horizontal eye movements and muscle tone (both from the zygomaticus major muscles and upper trapezius muscles on the neck) directly from the EEG cap.

– Please show sleep variables (e.g. the durations for each sleep stage).

We have provided a summary table of sleep architecture in the revised version of the Supplementary Materials. The table shows descriptive statistics of sleep architecture on MDnight and Cnight. Also, we report the result of the paired comparison between the nights and the Spearman correlations between the deprivation indices (DI before and DI after) and the changes between the nights in sleep architecture. Tests indicate that MD does not produce any main effect on the sleep architecture and that there are no substantial associations found between sleep architecture parameters and deprivation indices. Thus, it appears that changes in SSO and spindle frequency and amplitude did not lead to an alteration in the amount of N2 or N3 sleep, as we might expect. At the beginning of the Results section we refer to the table and to the lack of statistically significant effects.

– Please indicate the experimental design for the other control sessions (CN and MDM) in a figure. I think it would be helpful for the readers if all the conditions are shown in one figure (e.g. in Figure 1A).

We have added two panels in Figure 1 and 2 showing the experimental paradigm of the Cnight and the Mdmorn sessions. Thank you.

– Please provide information as to approximately what time each phase (MD, binocular rivalry) occurred in MDN, CN, and MDM sessions (page 18).

Information about each phase time has been provided in the section Main Study of the Experimental procedures within Materials and methods. Thank you.

– Page 18, 9AM to 14 AM at the laboratory. Is this 14PM?

Corrected, thank you.

Reviewer #2 (Recommendations for the authors):My specific concerns and recommendations are as follows:1) Figure 1, B1-3 – can indicators of statistical test results be put into the figure, for clarity?

We have added asterisks indicating the level of significance for individual time points in Figure 1 and the outcome of the ANOVA in Figure 2. Thank you.

2) For control night data in Figure 1 B1-2 – Are some of the control timepoint values statistically different from 1? It looks like there might be a deviation that either reflects (more likely) circadian time, or otherwise reflects an effect of wake time?

In the control night session, overall, the DI does not significantly differ across the different times. Because of the lack of a main effect, we did not think it necessary to perform post-hoc tests. Based on the reviewers’ input, we performed some exploratory post-hoc analyses and found that the DI measured in the control night before sleep was significantly larger than 1. We now report this result in the manuscript. We believe that this transient change in ocular dominance might reflect either a circadian effect or a transient form of adaptation due to the repetition of the binocular rivalry test, as reported in Klink et al., (2010).

3) For the MD data – since there seems to be a continuous change from the "Before" timepoint over time, and this is not seen after dark exposure spent awake (b3), is this definitely an effect of sleep, or could it also be an effect of time? e.g. are the "after 2h" time points in B1 and B3 statistically different from one another? A related question is since the deprivation index is lower in B3, is there a statistically significant effect of sleep, rather than time at all. Are the "before" timepoints in B1 and B3 statistically different?

Thank you for raising this important issue. We now directly compared the effect of monocular deprivation and its decay after two hours in the sleep vs dark exposure condition (MDnight vs MDmorn). We now plot the results of the two conditions in the same graph (Figure 2). We found a significant interaction effect between the factors TIME (before and after) and CONDITION (MDnight and MDmorn), indicating a specific role of sleep in prolonging the decay of short-term monocular deprivation.

4) Is there quantification of the sleep architecture in both B1 and B3? This should be provided, as it would be helpful for seeing relationship (or lack thereof) between these measures and prior REM or NREM time.

We have provided a summary table of sleep architecture in the revised version of the Supplementary Materials. The table shows descriptive statistics of sleep architecture on MDnight and Cnight. Also, we report the result of the paired comparison between the nights and the Spearman correlations between the deprivation indices (DI before and DI after) and the changes between the nights in sleep architecture. Tests indicate that MD does not produce any main effect on the sleep architecture and that there are no substantial associations found between sleep architecture parameters and deprivation indices. Thus, it appears that changes in SSO and spindle frequency and amplitude did not lead to an alteration in the amount of N2 or N3 sleep, as we might expect. At the beginning of the Results section we refer to the table and to the lack of statistically significant effects.

5) A general comment on the measure – I don't think this would be considered "consolidation" by most definitions. Consolidation usually means retention, or enhancement of plastic changes initiated during learning/experience. There may be a slowing of the rate of decay (this would need to be statistically tested by comparing sleep and awake controls), but the effects of MD lessen over time in all cases. It doesn't seem that there is any long-term effect of MD – quite different from sleep-dependent consolidation that was reported in critical period cat and adult mouse V1.

We thank the reviewer for raising this important issue. We agree that the data show a substantial delay in the decay process of the MD effects after the removal of the patch. The present data indicate that specifically the sleep condition and not merely darkness would be responsible for the maintenance of the MD-induced effect during the night. Therefore, we gladly adhere to the request and propose to say that sleep stabilizes/maintains the effects of MD as long as sleep itself persists. We have revised the entire manuscript through the various sections to handle this important aspect and to consider that a classic correlate of memory consolidation during sleep (spindles density) also turns out to be associated with maintenance of the MD-induced ocular dominance effect.

Also, we now directly compare the effect of monocular deprivation and its decay after two hours in the sleep vs dark exposure condition (MDnight vs MDmorn). In order to improve the clarity of the results we now plot the results of the two conditions in the same graph. We found a significant interaction effect between the factors TIME (before and after) and CONDITION (MDnight and MDmorn), indicating a specific role of sleep in stabilizing the decay of short-term monocular deprivation.

6) What do the authors make of figure 2 B4, given that it is frontal rather than occipital and correlated with DI after rather than before? what does this relationship mean?

For the sake of clarity, we have separated the figure relating to associations with MD-induced eye dominance (new figure 3) from that relating to associations with the residual dominance after two hours of sleep (new figure 4). In figure 4, the prefrontal ROI is introduced because we assumed an involvement of extra visual structures in the maintenance effect after sleep.

7) For Figure 2 – this is definitely showing that there is a relationship between prior plasticity in wake, and subsequent NREM sleep parameters, but is anything related to the change in this index ACROSS sleep? Such a change would be a convincing indication that there is indeed any kind of plasticity (or even slowing of decay or prior changes) occurring in the circuit during sleep.

We thank the reviewer for stimulating us to investigate whether there are any NREM parameters whose change within the sleep cycle can be related to the degree of plasticity maintenance observed at the end of the two hours of sleep. For this aim, we (1) partitioned SSO and spindle events into tertiles according to their occurrence time, (2) estimated the average measures of events belonging to the first and last tertile, and considered the variation between tertiles as an estimate of the changes across sleep. We then tested whether there is a consistent relationship between measures of individual retained plasticity (DI after) and changes in SSO and sleep spindles across sleep. None of the parameters considered showed associations across sleep with the plasticity index (DI after). We report these results in the supplementary materials.

8) On page 10: "..the power was reduced in participants showing a lower response to MD. This result, together with the other σ power estimates, reinforces the idea behind a modulation of thalamocortical interaction as a homeostatic reaction to MD. " A more parsimonious interpretation (in light of the lack of additional change during sleep – over and above what is seen in MD alone) is that propagation of activity through the circuit changes if the wiring has previously been changed. So this statement is not really an obvious interpretation.

We agree with the Reviewer that any causal interpretation could be hazardous; thus we tone down:

“This result, together with the other σ power estimates, reinforces the idea behind a modulation of thalamocortical interaction associated with plastic changes occurred during the MD”

Reviewer #3 (Recommendations for the authors):1. As mentioned above, it would be good to clarify the hypothesis motivating this study (p. 4, e.g. "we expected to observe a relation between SSO, σ oscillations and plasticity in visual area") and the putative mechanisms the authors derived from their results (exposed in the Discussion). I do think linking this result with previous findings on active memory consolidation, involving hippocampal ripples, sleep spindles and cortical slow waves is interesting. However, I struggle to see how the same mechanisms could be involved in MD-related plasticity. From the literature presented by the authors, it seems this plasticity affects the primary visual cortex, so not the form of hippocampus-dependent memory typically benefiting from active consolidation. In addition, I do not see what would be replayed here and how sleep spindles, given what we know about their involvement in memory consolidation, would act to consolidate this form of plasticity.

We thank the reviewer for stimulating us to better illustrate the rationale behind the study in the Introduction section. Again we would like to stress that this form of homeostatic plasticity is consolidated if repeated across days, as we have shown for amblyopic eye. In principle this is a form of memory and consolidation and as such it may rely on similar hippocampal processing as demonstrated for other type of memory consolidation. If so it would generalize the putative role of sleep spindle in different types of memory and plasticity.

2. The authors present their results as a "consolidation" but I do not see what is the argument for consolidation. To me, it seems that what they show is a maintenance of the effect during sleep. But they do not show a modification of the memory trace (enhancement, transfer, abstraction) that is usually being referred to as "consolidation".

We thank the reviewer for raising this issue. We agree that the data show a substantial delay in the decay process of the MD effects after the removal of the patch. The present data indicate that specifically the sleep condition and not merely darkness would be responsible for the maintenance of the MD-induced effect during the night. Therefore, we gladly adhere to the request and propose to say that sleep stabilizes/maintains the effects of MD as long as sleep itself persists. We have revised the entire manuscript through the various sections to handle this important aspect and to consider that a classic correlate of memory consolidation during sleep (spindles density) also turns out to be associated with maintenance of the MD-induced ocular dominance effect.

3. Regarding the MD-related plastic change, I am wondering if the authors could expand on the direction of the effect found in the different conditions. In particular, in the control night session, why would they obtain the reverse effect (DI>1)? Since there is no MD done in the control session, should the DI be equal to 1?

In the control night session, overall, the DI does not significantly differ across the different times. Because of the lack of a main effect, we did not think it necessary to perform post-hoc tests. Based on the reviewers’ input, we performed some exploratory post-hoc analyses and found that the DI measured in the control night before sleep was significantly larger than 1. We now report this result in the manuscript. We believe that this transient change in ocular dominance might reflect either a circadian effect or a transient form of adaptation due to the repetition of the binocular rivalry test, as reported in Klink et al., (2010).

4. The authors used ROI to explore the relationship between MD-related plastic changes and sleep features. How were these ROI defined? Why using ROI and not analyses at the electrode level? The topographical maps of the correlation coefficients seem to show clusters of electrodes in occipital area so an analysis at the electrode level should provide similar results but would rely on less arbitrary settings.

The introduction of ROIs stems from the hypotheses of the work. Given low spatial resolution of the EEG and having literature evidence that MD alters activity over extensive visual areas, we hypothesized that MD may alter cortical activity in occipito-parietal areas (and not at specific electrodes) during sleep. Also, another ROI was identified in the prefrontal cortex for its key role in organizing the ripple-mediated information transfer from hippocampus during NREM sleep. ROI-based analysis is not only in line with the assumptions of the work but also allowed us to decrease the number of statistical tests (we evaluate 3 ROIs instead of 90 electrodes) and made it possible to manage inter-individual variability of brain structures, in particular the large anatomical variability of V1 orientation implying a variably oriented dipole and a variable maximal representation of visual potentials over electrodes from Oz to CPz. In the Statistical analyses section of Materials and methods we have provided reasons to support this choice.

With these limitations in mind, we very gladly adhere to the reviewer's request to evaluate the effects on individual electrodes in more detail. To this end we have prepared supplementary figures which show boxplots and scatterplots for the electrodes inside the ROIs to evaluate main effects and associations, respectively

5. To correct multiple comparisons, the authors used the FDR method. But it is unclear how this was applied. How were groups of multiple tests defined? If the number of tests is low, the FDR is not, I think, the best approach to correct for multiple comparisons. It is typically used when correcting for hundreds or thousands of tests (e.g. in fMRI).

We thank the reviewer for highlighting an unclear point. In the revised version of the Statistical analyses section, we have provided missing details of the procedure used for handling false positives due to multiple testing. Basically, we applied the FDR correction for each question we asked. For example, “at which time points does dominance remain significantly different from baseline?” or, “which EEG feature and in which area of the scalp shows changes significantly dependent on plasticity induced by monocular deprivation?” For each of these questions, we made a group of tests (for the first example, dependent on the number of points at which ocular dominance was assessed until the morning; for the second example, on the number of EEG features examined multiplied by the number of areas in which they were assessed, i.e. up to 24) to which Benjamini and Hochberg's FDR correction was then applied.